# The mitochondrial β-oxidation enzyme HADHA restrains hepatic glucagon response by promoting β-hydroxybutyrate production

An Pan[1], Xiao-Meng Sun[1], Feng-Qing Huang[1], Jin-Feng Liu ![ORCID][2], Yuan-Yuan Cai[1], Xin Wu[1], Raphael N. Alolga[2], Ping Li[1], Bao-Lin Liu[1], Qun Liu ![ORCID][1,2✉] & Lian-Wen Qi ![ORCID][1,2✉]

Disordered hepatic glucagon response contributes to hyperglycemia in diabetes. The regulators involved in glucagon response are less understood. This work aims to investigate the roles of mitochondrial β-oxidation enzyme HADHA and its downstream ketone bodies in hepatic glucagon response. Here we show that glucagon challenge impairs expression of HADHA. Liver-specific HADHA overexpression reversed hepatic gluconeogenesis in mice, while HADHA knockdown augmented glucagon response. Stable isotope tracing shows that HADHA promotes ketone body production via β-oxidation. The ketone body β-hydroxybutyrate (BHB) but not acetoacetate suppresses gluconeogenesis by selectively inhibiting HDAC7 activity via interaction with Glu543 site to facilitate FOXO1 nuclear exclusion. In HFD-fed mice, HADHA overexpression improved metabolic disorders, and these effects are abrogated by knockdown of BHB-producing enzyme. In conclusion, BHB is responsible for the inhibitory effect of HADHA on hepatic glucagon response, suggesting that HADHA activation or BHB elevation by pharmacological intervention hold promise in treating diabetes.

---

[1] State Key Laboratory of Natural Medicines, School of Traditional Chinese Pharmacy, China Pharmaceutical University, Nanjing 210009, China. [2] Clinical Metabolomics Center, China Pharmaceutical University, Nanjing 211198, China. ✉email: liuquncpu@126.com; Qilw@cpu.edu.cn

Glucose homeostasis is regulated by hormones and metabolic intermediators. Insulin promotes glucose disposal after a meal, while glucagon mediates hepatic gluconeogenesis to restore euglycemia during starvation. When the liver and muscle stores of glycogen are depleted, ketone bodies produced in the liver supply energy substrates to metabolically active tissues, such as the heart and brain[1]. Ketogenic substrates include fatty acids and amino acids, and the acetyl-CoA substrate pool for ketogenesis mainly derives from fatty acid oxidation[2]. Hyperglucagonemia, a known trait of patients with diabetes, is responsible for the hyperglycemia[3]. Enhanced hepatic glucagon response induces cellular stress and disturbs metabolism in metabolic disorders[4,5], though the direct impact of glucagon on ketogenesis remains to be fully understood.

The shift from glucose metabolism to fat and ketone metabolism is a flexible metabolic mechanism that uses free fatty acids as the main energy source to minimize glucose utilization. In the fasted state, fat mobilization increases hepatic fatty acid uptake, and mitochondrial β-oxidation provides acetyl-CoA. Mitochondrial 3-hydroxy-3-methylglutaryl-CoA synthase 2 (HMGCS2) condenses acetyl-CoA with acetoacetyl-CoA to form HMG-CoA. HMG-CoA is then converted to acetoacetate (AcAc) by HMG-CoA lyase. At this point, AcAc can be further converted to β-hydroxybutyrate (BHB) by β-hydroxybutyrate dehydrogenase 1 (BDH1) in the liver[6]. BHB, AcAc and acetone are generally termed as ketone bodies, and BHB is the most abundant of the three[6]. Mitochondrial trifunctional protein (MTP) catalyzes three reactions in the fatty acid β-oxidation, and acetyl-CoA is liberated therefrom. MTP is composed of four alpha and four beta subunits encoded by the HADHA and HADHB genes, respectively[7]. MTP deficiency in mice is reported to result in nonalcoholic fatty liver disease[7]. Suppression of HADHA is shown to induce hepatic lipid deposition with hyperglycemia[8]. Consistently, loss of HMGCS2 in chow-fed mice causes mild hyperglycemia and increases hepatic gluconeogenesis[9]. These events indicate the potential role of fat oxidation and ketogenesis in hepatic glucose homeostasis.

Aside from its role as a passive carrier of energy, BHB also acts as a signaling metabolite[6]. By binding to membrane receptors, GPR109 and GPR41, BHB reduces lipolysis and sympathetic tone, and thus lowers metabolic rate[10–12]. BHB is structurally similar to butyrate, a canonical histone deacetylases (HDAC) inhibitor. BHB has been identified to inhibit Class I HDACs that deacetylate lysine residues on histone and non-histone protein to regulate genes encoding oxidative stress[13]. Acetylation modification is also involved in gluconeogenesis and mitochondrial function[14–16]. In addition, BHB is demonstrated to inhibit NLRP3 inflammasome-mediated inflammation[17].

In addition to driving glucose production, glucagon has been proposed to increase fatty acid oxidation and the production of ketone bodies[18]. Glucagon increases BHB production when perfused with oleic acid in rat liver[19]. In patients with type 1 diabetes, suppression of glucagon secretion by somatostatin prevented ketoacidosis after acute insulin withdrawal[20]. However, this view is challenged by a recent study, which showed that blocking glucagon receptor lowered fasting and fed glycemic levels, but did not alter BHB levels, suggesting that glucagon receptor signaling is not essential for ketogenesis[21]. In fact, the relationships among glucagon action, fatty acid oxidation and ketogenesis are very complex and involve diverse metabolic pathways, thus, making it difficult to precisely distinguish the sequence of altered metabolism from observed phenotypes in vivo.

This work aimed to characterize the role of HADHA and its downstream ketone bodies in hepatic glucagon response. We showed here that glucagon challenge impaired HADHA and downstream BHB production, leading to excessive hepatic glucose output. The ketone body, BHB, selectively inhibited Class IIa HDAC7 to inactive FOXO1 via preserving acetylation. Our findings present BHB as a negative regulator of excessive hepatic glucose production in diabetes.

## Results

**Glucagon impaired HADHA to promote gluconeogenesis.** To mimic hyperglucagonemia in diabetes, we established in vitro and in vivo models using high doses of glucagon (100 nM for primary hepatocytes and 2 mg/kg for mice). Glucagon impaired HADHA protein abundance (Supplementary Fig. 1a) but had no effect on HADHB (Supplementary Fig. 1b) in primary hepatocytes. In close agreement, a significant decline in hepatic HADHA expression was observed in the acute glucagon-challenged mice (Fig. 1a). The decreased level of HADHA was replicated in high-fat diet (HFD)-fed mice and ob/ob mice compared with normal control (Supplementary Fig. 1c). Since the mRNA of HADHA was not affected by glucagon (Supplementary Fig. 1d), we investigated whether glucagon impaired HADHA induction via post-transcriptional regulation. In hepatocytes, when protein synthesis was inhibited by cycloheximide, glucagon promoted HADHA protein degradation rapidly from 0 min to 60 min (Supplementary Fig. 1e). Lysosomal inhibitor chloroquine, but not proteasome inhibitor MG132, prevented the loss of HADHA protein, suggesting that glucagon possibly impaired HADHA protein stability via lysosomal degradation (Supplementary Fig. 1f).

To investigate the potential implication of HADHA in gluconeogenesis, we used AAV8-shRNA to establish liver-specific HADHA knockdown mice (Supplementary Fig. 2a). Hepatic HADHA deficiency alone is sufficient to increase fasting glucose in mice (Fig. 1b). In glucagon tolerance test, hepatic HADHA deficiency augmented hyperglycemic response (Fig. 1c). Expressions of the hepatic gluconeogenic genes, Pck1, G6pc and Pgc1a were enhanced by HADHA knockdown in the absence and presence of glucagon challenge (Fig. 1d). In contrast, liver-specific HADHA overexpression (Supplementary Fig. 2b) reduced fasting blood glucose (Fig. 1e), improved glucagon tolerance (Fig. 1f) and inhibited gluconeogenic genes expression in mice challenged with or without glucagon (Fig. 1g). Similar to what was observed in vivo, HADHA knockdown (Supplementary Fig. 3a) augmented glucagon-induced gluconeogenic genes expression (Supplementary Fig. 3b) and hepatic glucagon production (Supplementary Fig. 3c) in primary hepatocytes. Conversely, HADHA overexpression (Supplementary Fig. 3d) antagonized glucagon-stimulated hepatic gluconeogenesis (Supplementary Fig. 3e, f).

Glucagon is shown to stimulate insulin secretion, and insulin could also reduce hepatic glucose output[22]. Glucagon challenge exhibited the potential to stimulate insulin secretion (Supplementary Fig. 4a). Somatostatin, an inhibitor of pituitary growth hormone secretion, effectively reduced insulin release (Supplementary Fig. 4a). But somatostatin failed to reverse the inhibitory effects of HADHA on hepatic glucagon responses (Supplementary Fig. 4b, c), suggesting HADHA suppressed gluconeogenesis independently of insulin secretion. In addition, HADHA knockdown did not impair hepatic insulin sensitivity in the absence or presence of glucagon in mice (Supplementary Fig. 4d).

**Isotope tracing revealed glucagon inhibited ketogenesis by suppressing HADHA.** HADHA is involved in β-oxidation to generate acetyl-CoA, a major precursor of ketogenesis and the tricarboxylic acid (TCA) cycle. Stable isotope tracing combined with mass spectrometry analysis were employed to study the effects of glucagon and HADHA in ketogenesis (Fig. 2a). [U-$^{13}$C] palmitate at 0.1 mM was incubated with primary hepatocytes for 4 h. Analysis of both labeled (M2) and unlabeled (M0) metabolite fractions showed that glucagon stimulation slightly inhibited

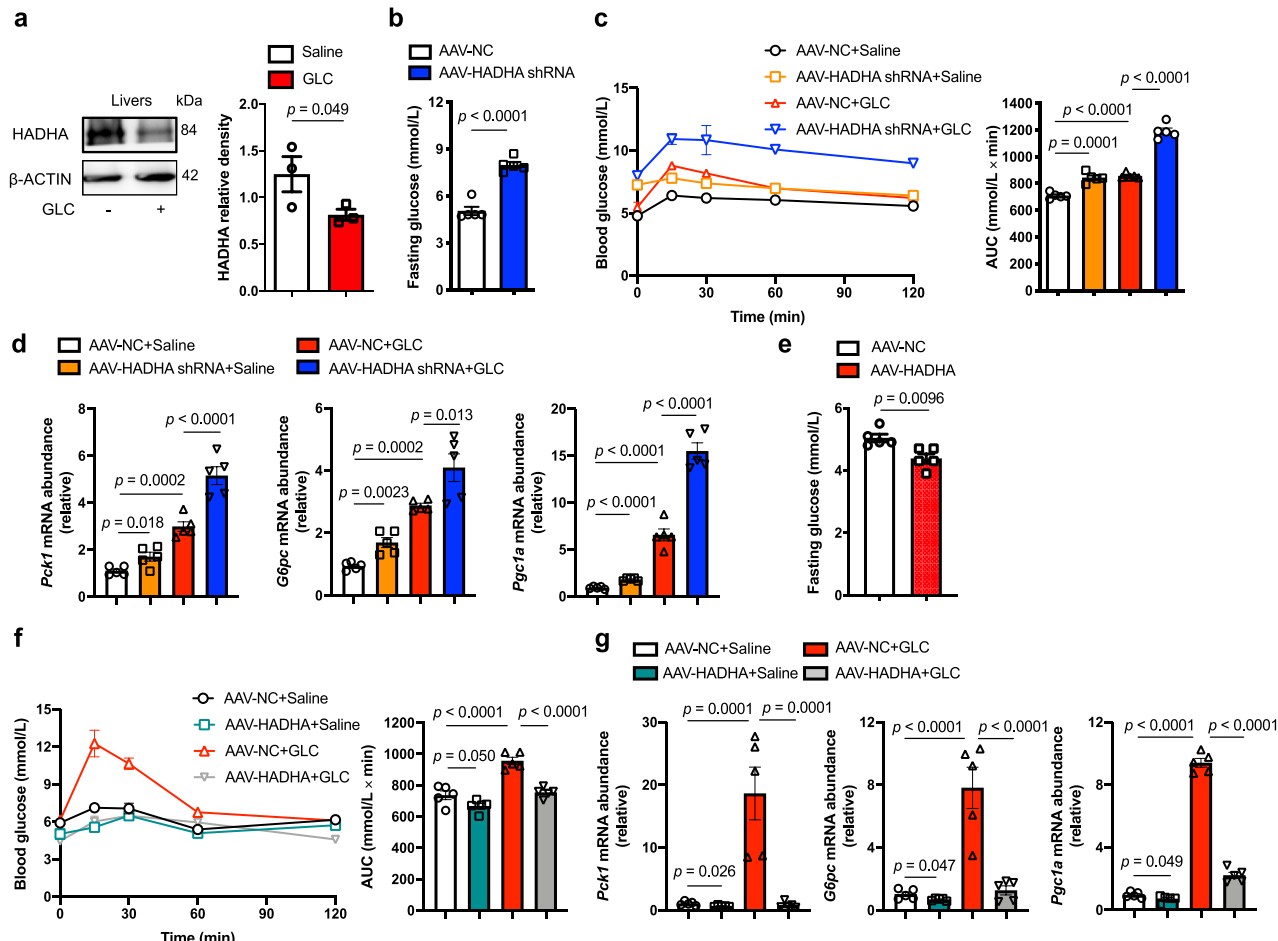

**Fig. 1 HADHA inhibited glucagon-stimulated hepatic gluconeogenesis. a** Hepatic HADHA protein level in glucagon-challenged mice (2 mg/kg, $n = 3$). **b** Fasting blood glucose of liver-specific HADHA knockdown mice ($n = 5$). **c** Blood glucose levels in normal mice subjected to glucagon challenge (2 mg/kg) and treated with AAV8-HADHA shRNA or AAV8-NC. AUC is indicated on the right ($n = 5$). **d** mRNA levels of *Pck1*, *G6pc* and *Pgc1a* in the livers of the mice in **c** ($n = 5$). **e** Fasting blood glucose of liver-specific HADHA-overexpressed mice ($n = 5$). **f** Blood glucose curve and AUC for mice treated with AAV8-HADHA or AAV8-NC after glucagon injection (2 mg/kg, $n = 5$). **g** mRNA levels of *Pck1*, *G6pc* and *Pgc1a* in the livers of the mice in **f** ($n = 5$). AAV adeno-associated virus, AUC area under the curve, GLC glucagon, NC normal control. Bars represent mean ± SEM values. Statistical differences between two groups were determined by a two-tailed Student's $t$ test, and all others were used by one-way ANOVA. Source data are provided as a Source Data file.

acetyl-CoA (Supplementary Fig. 5a). HADHA overexpression considerably enhanced acetyl-CoA generation, and restored acetyl-CoA production that was inhibited by glucagon (Supplementary Fig. 5a). Elisa analysis confirmed that HADHA overexpression increased the levels of acetyl-CoA in the absence or presence of glucagon (Supplementary Fig. 5b).

The ketone bodies are mainly formed from β-oxidation-derived acetyl-CoA, and BHB and AcAc account for >95%[6]. Glucagon stimulation significantly inhibited BHB (Fig. 2b) and AcAc (Fig. 2c) in labeled (M2 and M4) and unlabeled (M0) fractions. HADHA overexpression promoted BHB and AcAc generation in the absence or presence of glucagon (Supplementary Fig. 5b and Fig. 2b, c). In addition, glucagon stimulation inhibited the metabolite citrate (Cit), a TCA cycle intermediate (Fig. 2d). HADHA overexpression promoted Cit generation (Fig. 2d). Conversely, HADHA knockdown inhibited production of BHB, AcAc and Cit, and further aggravated the inhibitory effects of glucagon in ketogenesis (Fig. 2e–g).

In line with in vitro results by isotope tracing, the serum BHB level was significantly decreased by glucagon stimulation in mice (Supplementary Fig. 6a). Hepatic HADHA overexpression alone increased serum BHB level, and restored serum BHB level that was inhibited by glucagon stimulation (Supplementary Fig. 6b).

Conversely, HADHA knockdown alone decreased BHB level, and further aggravated the inhibitory effects of glucagon on BHB production (Supplementary Fig. 6c).

**BHB mediated the inhibitory effects of HADHA on hepatic gluconeogenesis.** We have shown that glucagon downregulated HADHA to decrease ketogenesis, and HADHA was a negative regulator of hepatic gluconeogenesis. We thus proposed a hypothesis that ketone bodies mediated the inhibitory effects of HADHA on glucagon response. Intraperitoneal injection of BHB (100 mg/kg, Supplementary Fig. 7) in mice attenuated hyperglycemic response in glucagon tolerance test (Fig. 3a) and correspondingly inhibited hepatic gene expressions of *Pck1*, *G6pc* and *Pgc1a* (Fig. 3b). Ketone bodies were shown to induce insulin secretion[23,24]. When the insulin level was reduced by somatostatin (Supplementary Fig. 8a), the inhibitory effect of BHB on gluconeogenesis was not altered (Supplementary Fig. 8b, c), suggesting that suppression of gluconeogenesis by BHB is independent of insulin. In isolated primary hepatocytes, BHB pretreatment inhibited glucagon-induced gene expressions of *Pck1*, *G6pc* and *Pgc1a* in a dose-dependent manner (Supplementary Fig. 9). BHB at 400 µM effectively inhibited hepatic gluconeogenesis (Fig. 3c, d) and was selected for further in vitro assays.

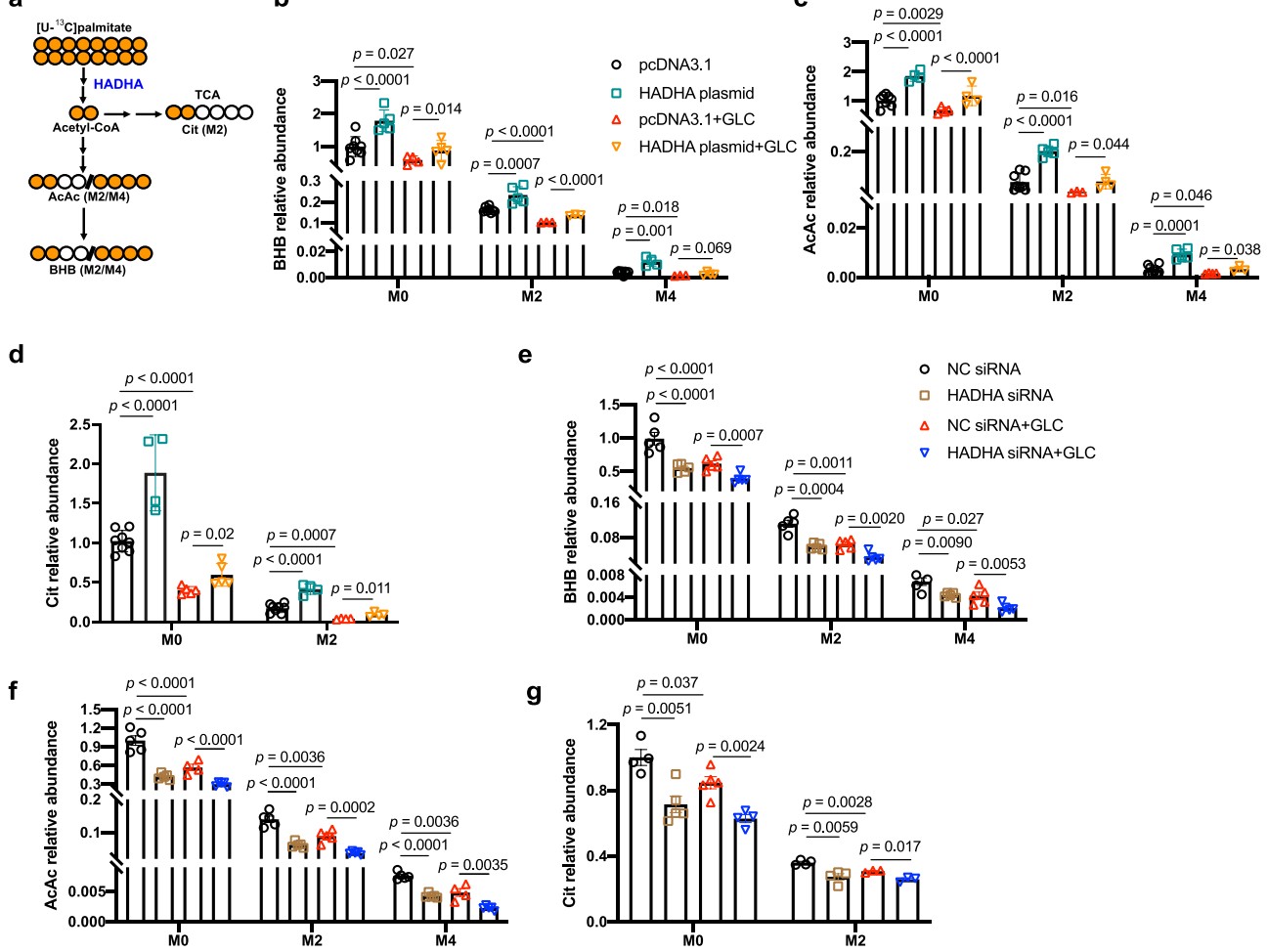

**Fig. 2 Stable isotope tracing of ketogenesis. a** A schematic summary of isotope tracing of ketogenesis using [U-$^{13}$C]palmitate. **b–d** Primary hepatocytes treated with [U-$^{13}$C]palmitate (0.1 mM) for 4 h followed by 100 nM glucagon stimulation for 1 h with or without HADHA plasmid transfection. **b** BHB relative abundance. $n = 8, 5, 4, 5, 8, 5, 3, 3, 8, 4, 3, 4$ (from left to right). **c** AcAc relative abundance. $n = 8, 4, 4, 4, 8, 5, 3, 4, 8, 5, 4, 3$ (from left to right). **d** Cit relative abundance. $n = 8, 4, 5, 5, 8, 4, 4, 4$ (from left to right). **e–g** Primary hepatocytes treated with [U-$^{13}$C]palmitate (0.1 mM) for 4 h followed by 100 nM glucagon stimulation for 1 h with or without HADHA siRNA transfection ($n = 3$–5). **e** BHB relative abundance. $n = 5, 5, 5, 5, 5, 5, 5, 5, 4, 5, 5, 5$ (from left to right). **f** AcAc relative abundance $n = 5, 5, 4, 5, 5, 5, 5, 5, 5, 5, 4, 5$ (from left to right). **g** Cit relative abundance. $n = 4, 5, 5, 4, 4, 4, 3, 3$ (from left to right). AcAc acetoacetate, BHB β-hydroxybutyrate, Cit citrate, GLC glucagon, TCA tricarboxylic acid cycle. Bars represent mean ± SEM values. Statistical differences were determined by one-way ANOVA. Source data are provided as a Source Data file.

Exogenous BHB inhibited HADHA knockdown-enhanced glucagon response (Supplementary Fig. 10a, b). Silencing BDH1 (Supplementary Fig. 11a), the last enzyme of hepatic ketogenesis that converts AcAc to BHB[25], antagonized HADHA-induced BHB production in hepatocytes (Supplementary Fig. 11b). The inhibitory effect of HADHA overexpression on glucagon-induced gluconeogenesis was abolished in BDH1 deficient hepatocytes, and was rescued by pretreatment with BHB (Fig. 3e, f). These findings suggest that BHB is responsible for the inhibitory effects of HADHA on hepatic gluconeogenesis.

We next studied the impact of another ketone body, AcAc, on gluconeogenesis. AcAc showed similar effects to BHB in inhibiting gluconeogenic genes expression (Fig. 3g) and hepatic glucose production (Fig. 3h) in primary hepatocytes. However, BDH1 knockdown diminished the inhibitory effects of AcAc in glucagon-induced gluconeogenesis (Fig. 3g, h), suggesting AcAc inhibited glucagon response in a BDH1-dependent manner. It is presumed that AcAc inhibited glucagon response through being converted into BHB. To address this point, we knocked down BDH1 in hepatocytes to prevent interconversion of the two ketones, and then added BHB and AcAc separately. Interestingly,

BHB could still effectively inhibit glucagon-induced gluconeogenesis, but AcAc failed to produce same effect in BDH1-knockdown hepatocytes (Supplementary Fig. 12). Hence, BHB, but not AcAc, is responsible for the inhibitory effects of HADHA on hepatic gluconeogenesis.

**HADHA and BHB inhibited glucagon-stimulated FOXO1 activation**. In response to glucagon signaling, FOXO1 translocates to the nucleus, where it transcriptionally promotes gene induction of *Pck1* and *G6pc* in cooperation with the co-activator *Pgc1a*[26,27]. In FOXO1-enforced hepatocytes, HADHA overexpression reduced luciferase activity of *Pck1* and *G6pc* promoters (Fig. 4a), suggesting that HADHA blocked the transcriptional activity of FOXO1. Immunofluorescence image showed that HADHA overexpression promoted FOXO1 nuclear exclusion, an effect that was diminished in BDH1 knockdown hepatocytes (Fig. 4b). As expected, BHB also prevented nuclear translocation of FOXO1 in response to glucagon treatment (Fig. 4c).

Nuclear exclusion of FOXO1 is regulated by its phosphorylation. HADHA overexpression preserved FOXO1 phosphorylation in

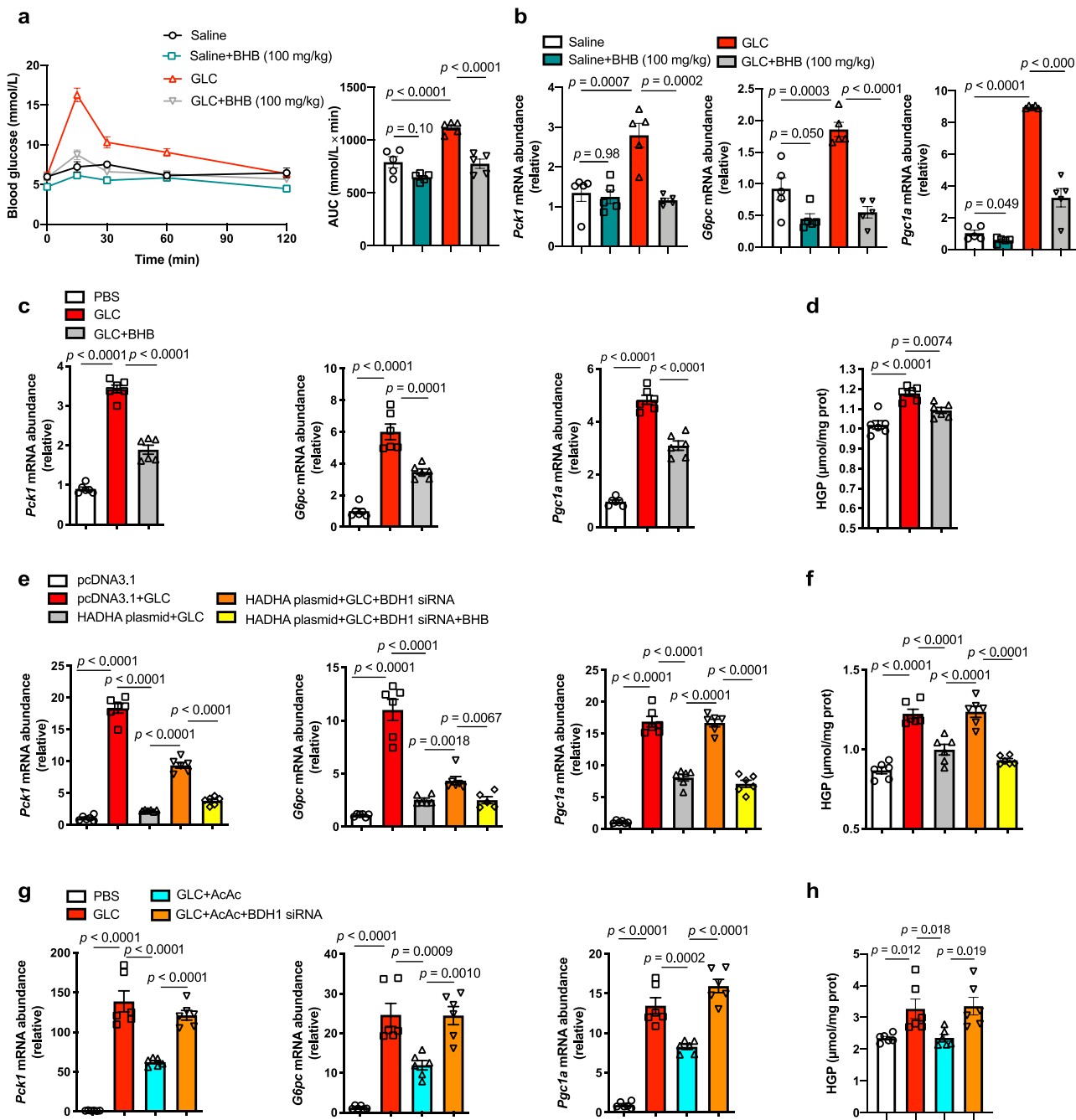

**Fig. 3 BHB was responsible for the inhibitory effect of HADHA on hepatic gluconeogenesis. a** Blood glucose levels in normal mice subjected to glucagon challenge (2 mg/kg) and those with BHB treatment (100 mg/kg). AUC is indicated on the right (n = 5). **b** mRNA levels of Pck1, G6pc and Pgc1a in the livers of the mice in **a** (n = 5). **c** mRNA abundance of Pck1, G6pc and Pgc1a in glucagon-stimulated (100 nM, 1 h) primary hepatocytes treated with or without BHB (400 μM, 6 h, n = 5). **d** HGP from the hepatocytes in **c** (n = 6). **e** Relative mRNA abundance of Pck1, G6pc and Pgc1a in HADHA-overexpressed primary hepatocytes transfected BDH1 siRNA with or without BHB pretreatment (400 μM, 6 h), 100 nM glucagon stimulation for 1 h (n = 6). **f** HGP from the hepatocytes in panel **e** (n = 6). **g** Relative mRNA abundance of Pck1, G6pc and Pgc1a in AcAc pretreated (400 μM) primary hepatocytes with or without BDH1 siRNA transfection (n = 6). **h** HGP from the hepatocytes in **g** (n = 6). AcAc acetoacetate, AUC area under the curve, BHB β-hydroxybutyrate, GLC glucagon, HGP hepatic glucose production, PBS phosphate buffer saline. Values represent mean ± SEM. Statistical differences were determined by one-way ANOVA. Source data are provided as a Source Data file.

a BDH1-dependent manner (Fig. 4d), and BHB addition protected FOXO1 phosphorylation against glucagon (Fig. 4e). In contrast to phosphorylation regulation, deacetylation ensures the nuclear retention of FOXO1[28]. HADHA overexpression preserved FOXO1 acetylation in a BDH1-dependent manner (Fig. 4f). In contrast, HADHA knockdown potentiated glucagon-mediated FOXO1 deacetylation, which was prevented by BHB repletion

(Fig. 4g). These results suggest that HADHA and BHB promoted FOXO1 nuclear exclusion by enhancing its phosphorylation and acetylation.

**BHB selectively inhibited HDAC7 to preserve FOXO1 acetylation.** Acetylation makes FOXO1 more accessible for phosphorylation[16].

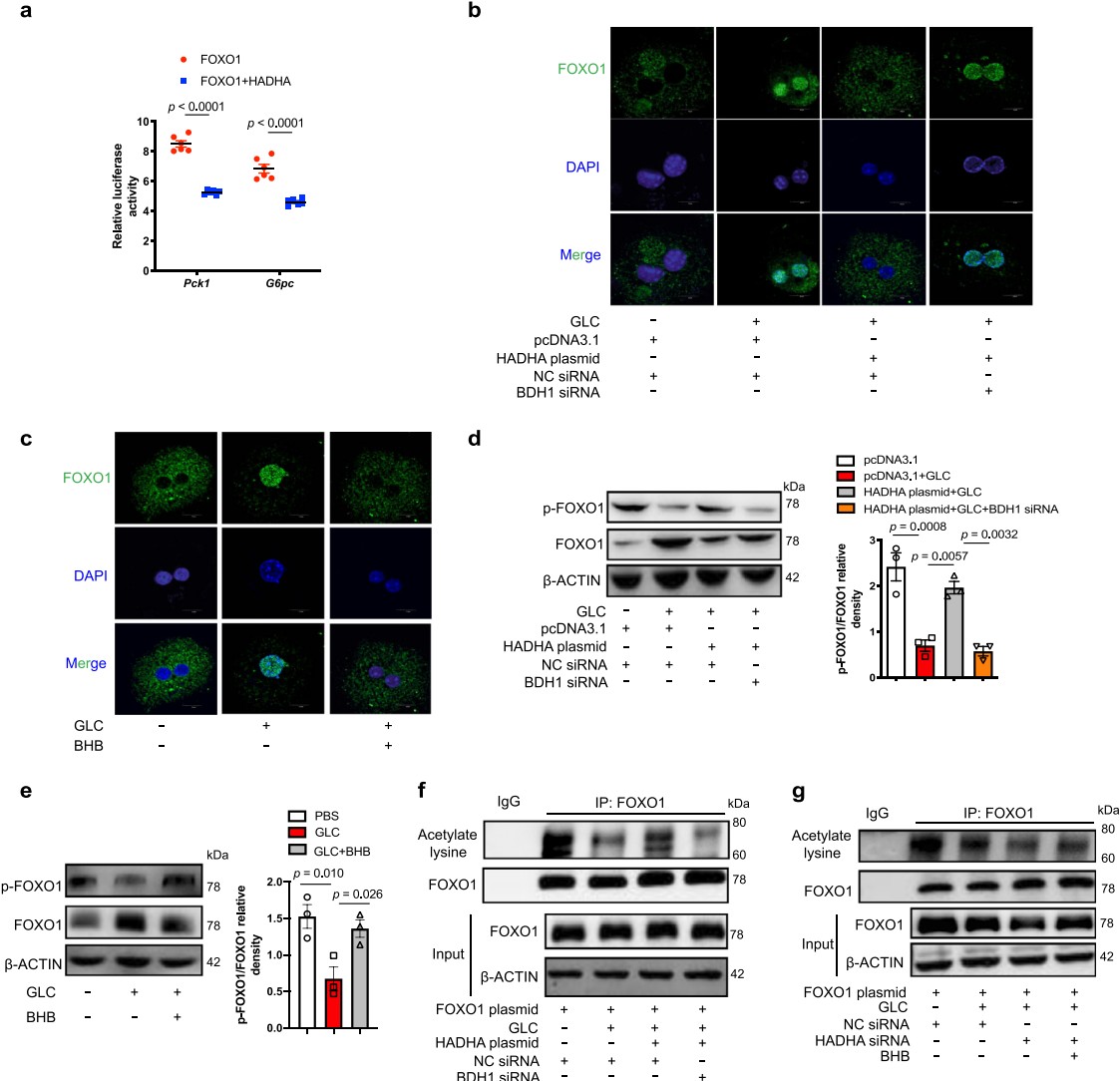

**Fig. 4 HADHA and BHB inhibited glucagon-induced FOXO1 activation. a** Luciferase reporter assay of the inhibitory effect of HADHA on *Pck1* and *G6pc* gene promoter co-transfected with HADHA and FOXO1 plasmid in HEK-293T cells. The luciferase activity was normalized with the internal control (Renilla luciferase, $n = 6$). **b** Representative confocal images of FOXO1 nuclear translocation in HADHA-overexpressed primary hepatocytes with or without BDH1 siRNA transfection and 100 nM glucagon stimulation for 1 h. Scale bar represents 10 μm. **c** Confocal images of FOXO1 nuclear translocation in BHB (400 μM, 6 h) treated primary hepatocytes when exposed to glucagon (100 nM, 1 h). Scale bar represents 10 μm. **d** FOXO1 phosphorylation from the hepatocytes in **b** ($n = 3$). **e** FOXO1 phosphorylation from the hepatocytes in panel **c** ($n = 3$). **f, g** FOXO1 acetylation in HADHA-overexpressed or knockdown primary hepatocytes transfected with BDH1 siRNA and 100 nM glucagon stimulation for 1 h. **b, c, f** and **g** were repeated 3 times independently with similar results. BHB β-hydroxybutyrate, GLC glucagon, NC normal control. Values represent mean ± SEM. Statistical differences were determined by one-way ANOVA. Source data are provided as a Source Data file.

Class IIa HDACs (HDAC4, 5 and 7) translocate to the nucleus to deacetylate FOXO1 for the induction of gluconeogenic genes[15]. Glucagon increased translocation of HDAC4, 5 and 7 to the nucleus (Fig. 5a). HDAC7, but not HDAC4 and 5, was restrained in the cytosol by HADHA overexpression in hepatocytes (Fig. 5a). The effect of HADHA on HDAC7 was diminished by BDH1 knockdown and then rescued by BHB repletion (Fig. 5a). The roles of HADHA and BHB in HDAC7 nuclear exclusion were further confirmed by immunofluorescence staining (Supplementary Fig. 13a–c). Immunoprecipitation assays showed that HADHA overexpression or BHB addition visibly reduced the binding of HDAC7 to FOXO1, but had no significant influence on the binding of HDAC4 and 5 to FOXO1 (Fig. 5b, Supplementary Fig. 13d). Glucagon increased enzymatic activities of HDACs, whereas BHB considerably inactivated HDAC7 without significant influence on HDAC4 and HDAC5 activity (Fig. 5c).

We further investigated the role of HDAC7 in the inhibitory effects of HADHA and BHB on hepatic glucagon response. HDAC7 overexpression (Supplementary Fig. 14a) diminished the inhibitory effects of HADHA or BHB on glucagon-induced expressions of *Pck1*, *G6pc* and *Pgc1a* in hepatocytes (Supplementary Fig. 14b, c). In contrast, HDAC7 knockdown (Supplementary Fig. 15) inhibited glucagon response and FOXO1 acetylation in the absence of BDH1 (Fig. 5d, e). Consistently, luciferase reporter showed that HDAC7 overexpression abrogated the inhibitory effects of HADHA on the activity of *G6pc* and *Pck1* promoters (Fig. 5f). Taken together, these data reveal that BHB inhibits enzymatic activity of HDAC7 to inactive FOXO1.

**The potential interaction between BHB and HDAC7.** Surface plasmon resonance (SPR) showed that BHB displayed a favorable

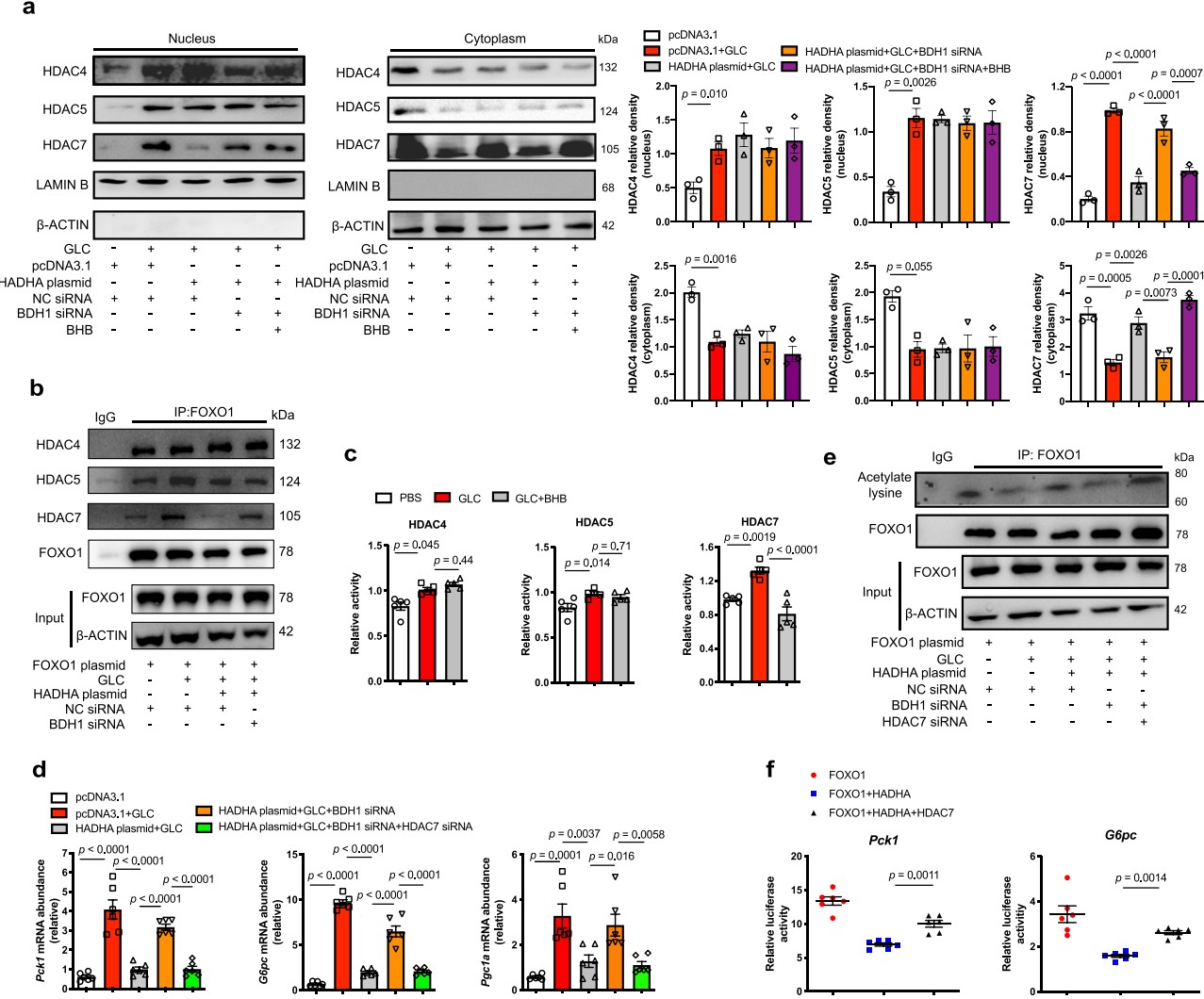

**Fig. 5 BHB selectively inhibited HDAC7 to preserve FOXO1 acetylation. a** The distribution of HDAC4, HDAC5 and HDAC7 in the nucleus and cytoplasm of HADHA-overexpressed hepatocytes transfected with BDH1 siRNA with or without BHB (400 µM, 6 h) and 100 nM glucagon stimulation for 1 h ($n = 3$). **b** Immunoprecipitation analysis of interaction of FOXO1 with HDAC4, HDAC5 or HDAC7 in HADHA-overexpressed hepatocytes with or without BDH1 siRNA transfection. It was repeated 3 times independently with similar results. **c** Enzyme activity of HDAC4, HDAC5 and HDAC7 in HepG2 cells with 100 nM glucagon stimulation for 1 h after BHB treatment (400 µM, 6 h, $n = 5$). **d** Relative mRNA abundance of *Pck1*, *G6pc* and *Pgc1a* in HADHA-overexpressed hepatocytes transfected with BDH1 siRNA and HDAC7 siRNA ($n = 6$). **e** Immunoprecipitation analysis of FOXO1 acetylation in **d**. It was repeated 3 times independently with similar results. **f** Luciferase reporter assay of the effect of HADHA and HDAC7 on *Pck1* and *G6pc* gene promoter. The *Pck1* and *G6pc* luciferase reporters were co-transfected with HADHA, FOXO1 and HDAC7 plasmid in HEK-293T cells. The luciferase activity was normalized with the internal control (Renilla luciferase, $n = 6$). Values represent mean ± SEM. Statistical differences were determined by one-way ANOVA. Source data are provided as a Source Data file.

binding affinity to HDAC7 with an estimated equilibrium dissociation constant of 0.02 µM, stronger than 4.47 µM for HDAC4 and 3.19 µM for HDAC5 (Fig. 6a). Cellular thermal shift assay was employed to confirm the BHB-HDAC7 interaction. We noted that BHB incubation stabilized HDAC7 at multiple temperatures assayed for the examined heat-denatured hepatocytes (Fig. 6b). Furthermore, we designed a BHB-biotin probe to investigate its interaction with HDAC7 by immunoblotting analysis. BHB-biotin probe effectively pulled HDAC7 protein out, and their binding was reduced by the addition of unlabeled BHB due to competitive binding (Fig. 6c). These data reveal the potential interaction between BHB and HDAC7.

The binding modes of BHB to HDAC7 were predicted by AutoDock. The top 5 binding poses for BHB to HDAC7 resided

exactly at the same pocket, which involved at least three hydrogen-bonds with His541, Glu543 and Asp626 (Fig. 6d). The docking scores for these top 5 binding poses were close to each other (−3.55 ~ −3.66 kcal/mol), as well as the root mean square deviations between their configurations (<0.2 Å). Other binding poses deviated significantly from the top 5 and showed less favorable docking scores. To further test the bindings of BHB to these three sites, alanine residues were introduced by site-directed mutagenesis to replace each site. Immunoblotting assay showed that Glu543 mutation, but not His541 and Asp626, impaired the binding of BHB-biotin to HDAC7 (Fig. 6e). Mutagenesis with alanine residue did not alter the HDAC7 enzyme activity (Fig. 6f). Compared with wild-type HDAC7, the inhibitory effects of BHB on *Pck1*, *G6pc* and *Pgc1a* were

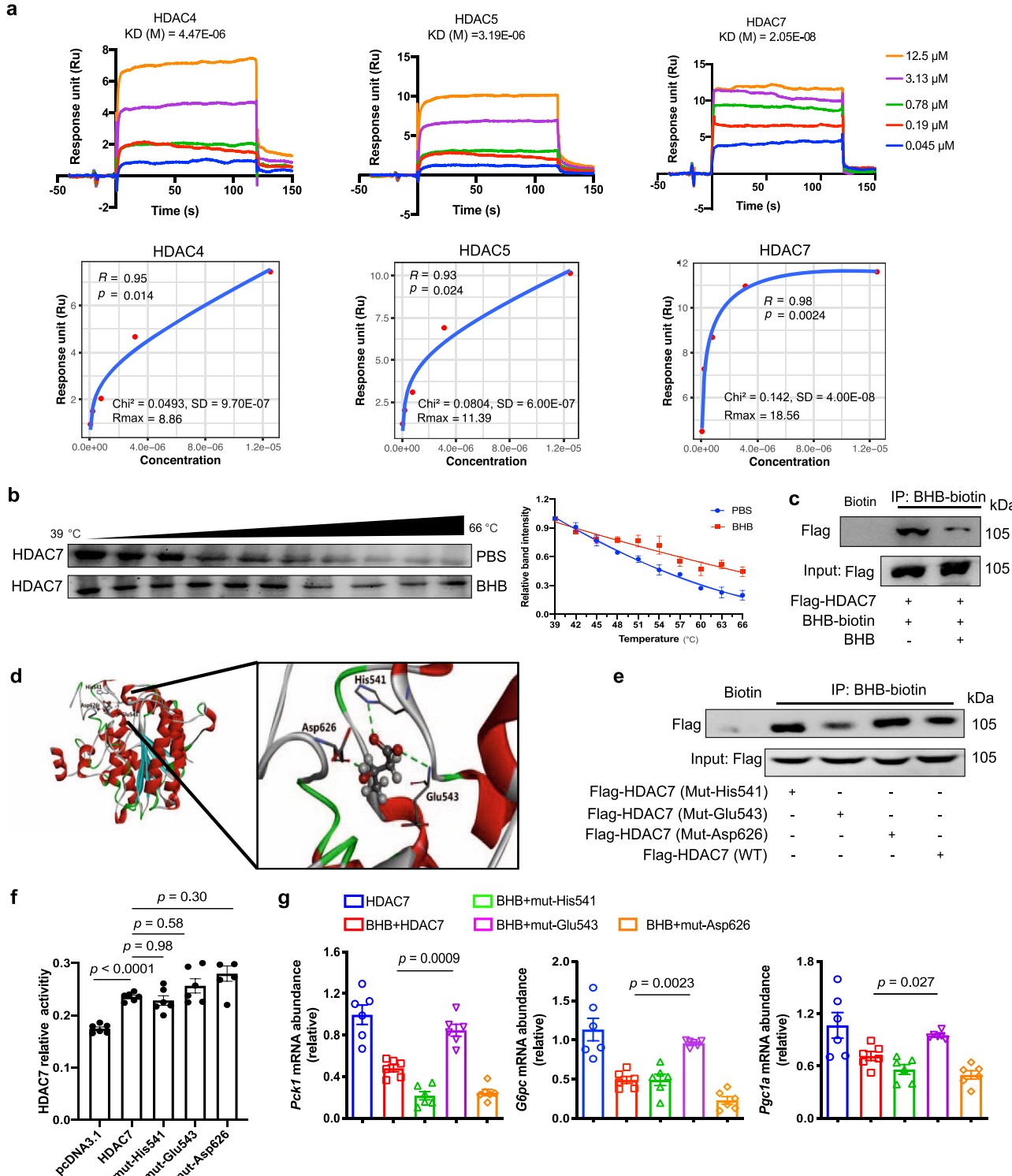

diminished by Glu543 mutation, but not His541 and Asp626 (Fig. 6g). Collectively, the data suggest that BHB binds to HDAC7 Glu543 to inhibit glucagon response.

**HADHA inhibited hepatic glucagon response in a BDH1-dependent manner in HFD-fed mice.** To confirm the negative role of HADHA and BHB in hepatic gluconeogenesis in vivo, hepatic HADHA was overexpressed by tail vein injection of AAV8 plasmid in HFD-fed mice (Supplementary Fig. 16a, b). HFD feeding for 12 weeks increased glucagon secretion (Fig. 7a).

The BHB levels in the serum (Fig. 7b) and the liver (Fig. 7c) were decreased by HFD feeding, which were normalized by HADHA overexpression (Fig. 7b, c). When hepatic BDH1 was silenced using AAV8-shRNA (Supplementary Fig. 16c, d), the role of HADHA in BHB production was blocked (Fig. 7b, c). Hepatic HADHA induction lowered fasting hyperglycemia and improved glucose and pyruvate tolerance with reduced hepatic lipid deposition in a BDH1-dependent manner (Fig. 7d–g). Meanwhile, weight gain and fat mass increased by HFD were also reduced by HADHA overexpression without affecting normal food intake (Supplementary Fig. 16e). HADHA induction

**Fig. 6 The potential interaction between BHB and HDAC7. a** The interaction of BHB with HDACs by surface plasmon resonance. A series of BHB concentrations was injected onto the HDAC4, HDAC5 or HDAC7 biosensor surface. The frequency response and fitting curves were displayed. **b** CETSA for in-cell HDAC7 target engagement. Representative western blots showed thermostable HDAC7 with indicated heat shocks in the presence or absence of BHB (50 μM, $n = 3$). **c** Immunoprecipitation analysis of interaction between biotin-BHB and HDAC7 in primary hepatocytes transfected with HDAC7 plasmid with or without unlabeled BHB treatment. It was repeated 3 times independently with similar results. **d** Binding modes and sites of BHB with HDAC7 predicted by AutoDock. The protein is shown as cartoon and BHB as sticks. **e** Immunoprecipitation analysis of interaction between BHB-biotin and HDAC7 in HEK-293T cells transfected with HDAC7 (WT, Mut-His541, Mut-Glu543 or Mut-Asp626) plasmid. It was repeated 3 times independently with similar results. **f** Enzyme activity of HDAC7 in HepG2 cells transfected with HDAC7 (WT, Mut-His541, Mut-Glu543 or Mut-Asp626) plasmid ($n = 5$). **g** Relative mRNA abundance of *Pck1*, *G6pc* and *Pgc1a* in primary hepatocytes transfected with HDAC7 (WT, Mut-His541, Mut-Glu543 or Mut-Asp626) plasmid after BHB treatment ($n = 6$). Asp aspartic acid, BHB β-hydroxybutyrate, CETSA, cellular thermal shift assay, GLC glucagon, Glu glutamic acid, His histidine, Mut mutant, WT wild type. Values represent mean ± SEM. Statistical differences were determined by one-way ANOVA. Source data are provided as a Source Data file.

improved metabolic parameters, but the effects were diminished by BDH1 deficiency (Supplementary Fig. 16f).

HFD feeding increased nuclear retention of HDACs, whereas HADHA overexpression promoted HDAC7 nuclear exclusion in a BDH1-dependent manner without affecting the distribution of HDAC4 and HDAC5 (Fig. 7h). HADHA induction preserved FOXO1 acetylation and inactivated FOXO1 by increasing phosphorylation, but this effect was diminished by BDH1 knockdown (Fig. 7i, j). As a consequence, HADHA inhibited gene induction of *Pck1, G6pc* and *Pgc1a* in a BDH1-dependent manner (Fig. 7k).

## Discussion
In this work, we characterized a critical role for HADHA and its downstream metabolite BHB in hepatic gluconeogenesis. The major findings of this work include: (1) Glucagon inhibits ketogenesis by suppressing HADHA, a key enzyme in fatty acid oxidation. (2) HADHA is identified as a negative regulator of hepatic gluconeogenesis. (3) BHB is responsible for HADHA-mediated hepatic gluconeogenesis inhibition by directly binding to the HDAC7 Glu543 site to preserve FOXO1 acetylation. The proposed mechanism is illustrated in Fig. 8. Our data point to a potential therapeutic strategy of targeting HADHA or the ketone body BHB for the treatment of diabetes.

Both impaired insulin action and aberrantly elevated glucagon are responsible for diabetic hyperglycemia[29]. Emerging studies have put a spotlight on the regulation of glucagon by β cells in the development of diabetes[30]. To mimic hyperglucagonemia in diabetes, we treated mice with high dose of glucagon (2 mg/kg)[21], and found impaired HADHA expression and ketone body production in the liver. These data are different from the traditional view that glucagon stimulates fatty acid oxidation and ketogenesis[18–20]. Of note, the fact that glucagon stimulates ketone body production was only observed when insulin signaling is blocked[21,31], but a direct relationship between glucagon and ketogenesis has not been well established. A possible explanation could be that besides elevated glucagon, insulin levels are low during prolonged fasting, facilitating adipose lipolysis and fatty acid availability for oxidation and ketone body generation. In line with this explanation, insulin receptor antagonist elevated serum BHB levels in fed mice, while blocking glucagon signaling had no effect on plasma BHB levels in insulin-sensitive mice[31]. Different from glucagon actions as a physiological response during fasting, hyperglucagonemia is a pathological insult in diabetes[27]. Similar to our observation in mice subjected to glucagon challenge, injection of glucagon at a supraphysiologic dose (1 mg/kg) is also shown to impair ketone body production in fasted mice[21]. Moreover, we showed the direct impact of glucagon in isolated hepatocytes. Glucagon challenge impaired HADHA protein to suppress β-oxidation, well explaining the reduction in ketogenesis under pathological conditions.

We identified BHB as a signaling metabolite that regulates gluconeogenesis by binding to HDAC7 to inactive FOXO1. Similarly, loss of HMGCS2, the rate-limiting enzyme in the ketogenesis pathway, was shown to increase hepatic gluconeogenesis in mice[9]. FOXO1, together with PGC-1α, increase the transcriptional regulation of the key gluconeogenic genes, and its activity is susceptible to acetylation modification[16,28]. FOXO1 shuttles between the cytosol and nucleus. AKT-dependent phosphorylation inactivates FOXO1 by promoting its nuclear exclusion[32]. Similar to FOXO1, Class IIa HDACs (HDAC4, 5, 7 and 9) function on a subcellular compartmental basis[16]. In response to glucagon, Class IIa HDACs translocate to the nucleus to enhance FOXO1 activity in the liver[15]. We found that BHB selectively inhibited HDAC7 and then prevent FOXO1 activation. Because acetylation sensitizes FOXO1 for phosphorylation by related activating kinases[28], the phosphorylation and nuclear exclusion of FOXO1 could be a means for BHB to suppress hepatic gluconeogenesis. BHB has been identified as a Class I HDAC inhibitor that regulates histone acetylation in the kidney cells[14]. Our findings further demonstrate that BHB regulates non-histone protein function via inactivating Class II HDAC7 in hepatocytes from the perspective of glucose and lipid metabolism. The binding cavity of HDAC7 is delimited by the amino acids His531, His541, Pro542, Glu543, Ile628, and Phe679[33]. The docked pose of BHB, structurally similar to the canonical HDAC inhibitor butyrate, shows interactions with Glu543[34]. Thus, the BHB molecule which is able to establish contacts with this unique pocket, can be a structural element addressing inhibition for this enzyme.

HFD feeding impaired hepatic HADHA and reduced BHB production. HADHA overexpression improved metabolic parameter, and this effect was blocked by hepatic BDH1 knockdown, suggesting a role of BHB in regulating metabolic disorders. Besides its action in hepatocytes, BHB can also be exported from the liver to extrahepatic tissues, where it possibly regulates lipid metabolism, inflammation and insulin resistance[10,11,17]. In HFD-fed mice, reduced β-oxidation shifts hepatic fatty acids to esterification, and diacylglycerol generated in this process can induce insulin resistance to increase hepatic glucose production[35].

It should be noted that glucagon secretion is also sensitive to amino acids and fatty acids. Amino acids stimulate glucagon secretion, likely by increasing the α cell number, and then glucagon secreted stimulates hepatic amino acid metabolism (ureagenesis), thereby establishing a mutual feedback cycle described as liver-α cell axis[36,37]. Fatty acids are shown to stimulate glucagon secretion from pancreatic islets[38], and activation of membrane G-protein receptor was proposed to be a reason[39]. Conversely, some clinical observation found that increased fatty acids suppressed glucagon secretion[40,41]. It remains to be explored regarding which metabolites contribute to or are more tightly coupled to glucagon secretion.

Our work has some limitations: (1) A ketogenic diet or BHB feeding is required for further in vivo experiments to validate the

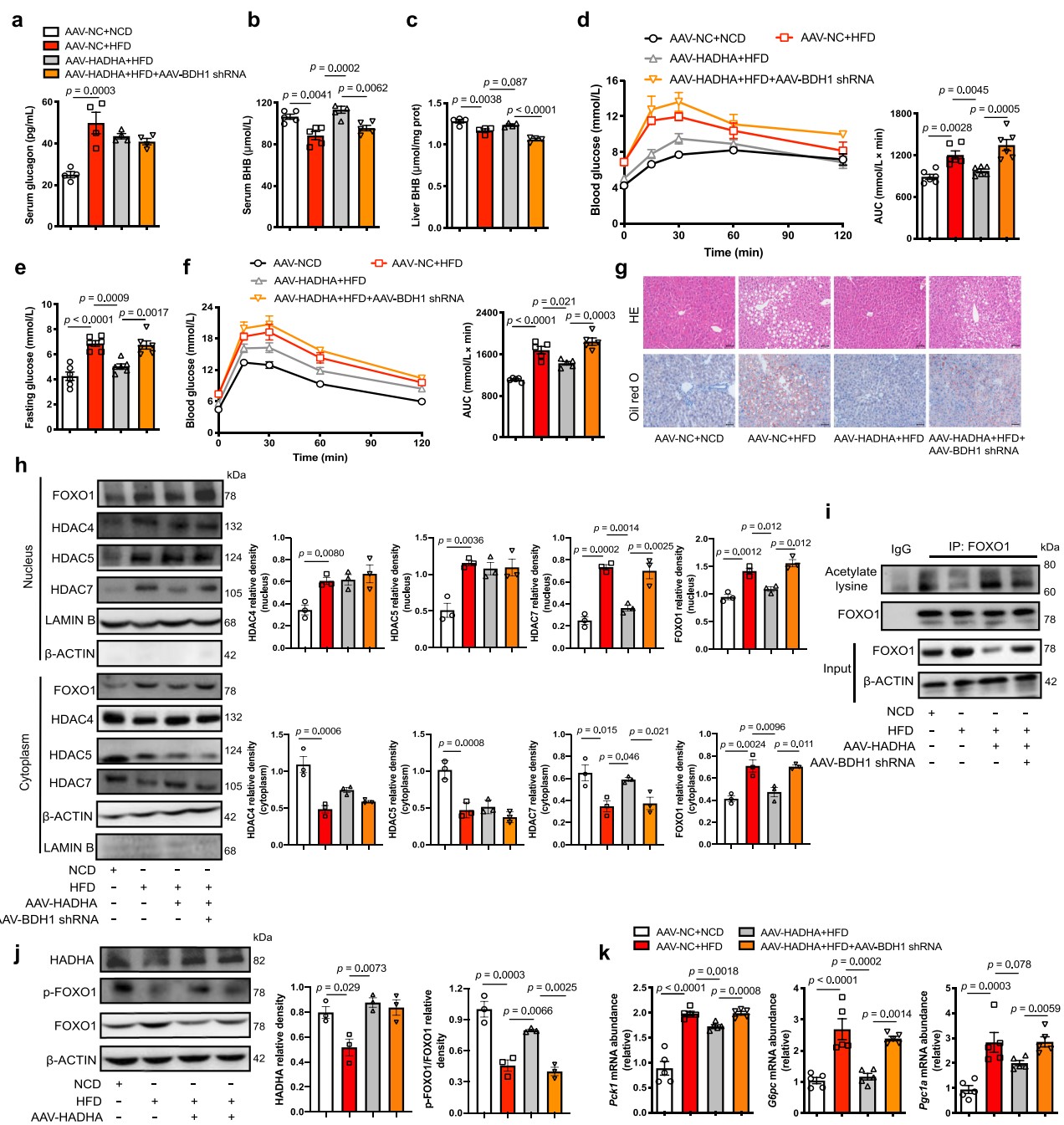

**Fig. 7 HADHA suppressed hepatic glucagon response dependent on BDH1 in HFD-fed mice.** Eight-week-old mice injected with AAV8-NC, AAV8-HADHA or AAV8-BDH1 shRNA were fed with NCD or HFD for 12 weeks. **a** Serum glucagon contents in mice ($n = 4$). **b, c** BHB contents in serum and liver (**b**: $n = 5$; **c**: $n = 4$). **d** Pyruvate tolerance test (2 g/kg body weight) after 11 weeks of feeding. AUC is indicated on the right ($n = 6$). **e** Fasting blood glucose after 12 weeks of feeding ($n = 6$). **f** Oral glucose tolerance test (2.5 g/kg) after 12 weeks of feeding. AUC is indicated on the right ($n = 5$). **g** Representative images of HE and Oil Red O-stained liver sections ($n = 3$). **h** Protein expressions of FOXO1, HDAC4, HDAC5 and HDAC7 in nucleus and cytoplasm of liver ($n = 3$). **i** Hepatic FOXO1 acetylation levels. It was repeated 3 times independently with similar results. **j** Western blotting analysis of HADHA and FOXO1 phosphorylation in the liver ($n = 3$). **k** Relative mRNA levels of *Pck1*, *G6pc* and *Pgc1a* in the liver ($n = 5$). AAV adeno-associated virus, AUC area under the curve, BHB β-hydroxybutyrate, HFD high-fat diet, NCD normal chow diet. Values represent mean ± SEM. Statistical differences were determined by one-way ANOVA. Source data are provided as a Source Data file.

role of ketone body on hepatic gluconeogenesis. (2) Other enzymes involved in β-oxidation that might play roles in glucagon response remain to be investigated. (3) The crystal structure of HDAC7 with BHB, a direct means to decipher protein-ligand interaction remains a challenging feat.

In summary, this work identified a previously unrecognized mechanism of HADHA and its downstream ketone body BHB in regulating glucagon-induced hepatic gluconeogenesis through selectively interacting with HDAC7 to facilitate FOXO1 inactivation. Our findings suggest a probable pharmacological intervention

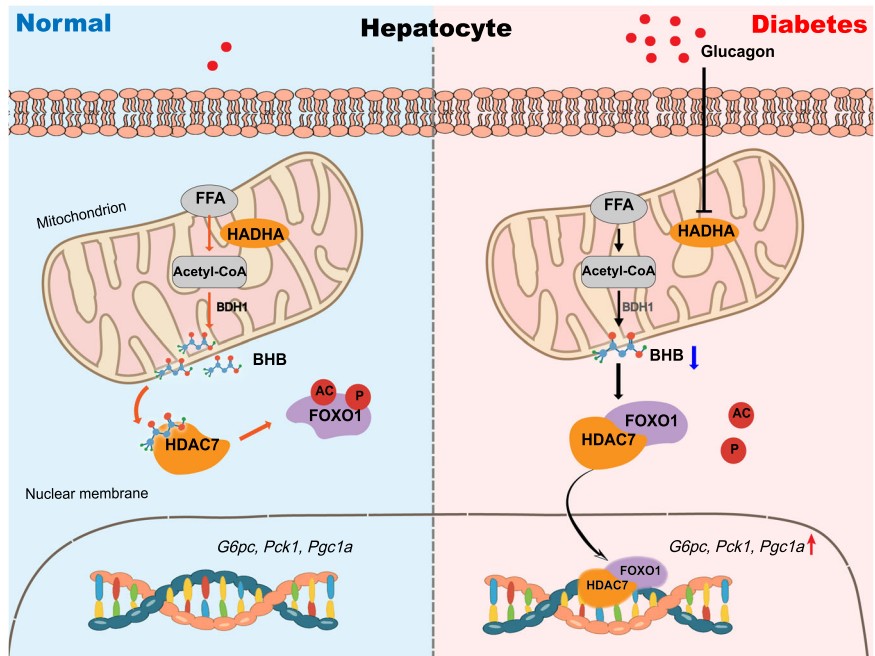

**Fig. 8 Scheme of the role for HADHA in promoting BHB production to restrain hepatic gluconeogenesis.** In normal state, hepatic HADHA facilitates ketogenesis from fatty acid β-oxidation. BHB, a predominant ketone body which is dependent on upstream HADHA expression, plays a key role in regulating gluconeogenic genes and hepatic glucose production. Mechanism-wise, BHB can bind to HDAC7 and inhibit its activity, thus preserving FOXO1 acetylation and phosphorylation. In diabetes, excess glucagon impairs HADHA expression and inhibits downstream BHB production, thereby increasing FOXO1 transcriptional activity and hepatic gluconeogenesis via HDAC7 activation.

of HADHA and dietary supplementation of BHB for the treatment of diabetes.

## Methods

**Animals and treatments**. All animal treatments were approved by the Animal Ethics Committee of China Pharmaceutical University (protocol no. 2019-04-002). Animal testing and research conformed to all relevant ethical regulations. Six to eight-week-old male C57BL/6J mice were purchased from Nanjing Junke Bio-technology Co, Ltd. (China). After a week of acclimation, the mice were raised in a temperature-controlled facility on a 12 h light-dark cycle with free access to food and water. For HFD feeding, mice were fed with HFD (60% kcal from fat; D12492; Research diet, America) or normal chow diet (10% kcal from fat; Xietong Organism, China) for 12 weeks.

For glucagon challenge experiments, the mice were intraperitoneally injected with 2 mg/kg glucagon (1 h, Novo Nordisk, Denmark). BHB (100 mg/kg, Aladdin, China) or somatostatin (3 mg/kg, Merck Serono, Switzerland) were administered to mice by intraperitoneal injection 1 h or 15 min before glucagon challenge.

**Liver-specific transfection in mice**. For liver-specific HADHA overexpression, mice were injected with 7 μl of AAV8-plasmid virus suspension (virus titer > 10¹³, Hanbio Biotechnology Co., Ltd, China) blended with 193 μl saline. Control mice were injected with AAV8-Normal Control (NC). Liver-specific knockdown of HADHA and BDH1 in mice were created using AAV8-shRNA technology, while mice for control were injected with AAV8-NC shRNA. Three weeks later, mice were randomly divided into groups. The shRNA oligo used in our experiments are as follows:

NC shRNA: TTCTCCGAACGTGTCACGT;
HADHA shRNA: GCTGACATCAACATGCTATCC;
BDH1 shRNA: GCAGGAAGTACU UCGAUGAAA.

**Glucagon, pyruvate and oral glucose tolerance tests in mice**. For glucagon and pyruvate tolerance tests, overnight fasted mice were administered with glucagon (2 mg/kg) or pyruvate (2 g/kg) by intraperitoneal injection. In the case of oral glucose tolerance test, fasted mice were orally administered with glucose (2.5 g/kg) by gavage. Tail-blood glucose levels were measured at 0, 15, 30, 60, 90, and 120 min after challenge using the ONETOUCH Ultra blood glucose meters (Johnson, America).

**Primary hepatocytes isolation and culture**. Primary hepatocytes were isolated from livers of male C57BL/6J mice. Briefly, liver cells were isolated by perfusion of

balanced salt solution (25 mM HEPES, 121 mM NaCl, 4.7 mM KCl, 1.2 mM MgSO₄, 5 mM NaHCO₃, 2 mM CaCl₂, 10 mM glucose, pH 7.4) containing 2% collagenase IV (Gibco, America) through the portal vein after anesthesia in mice. Dulbecco's modified eagle medium (DMEM) containing 50% Percoll (Sigma, America) was used to remove cell debris by gradient centrifugation. HepG2 cell line was obtained from the American Type Culture Collection, and human embryonic kidney 293 T (HEK-293T) cell line was obtained from Stem Cell Bank, Chinese Academy of Sciences. Primary hepatocytes, HepG2 cells and HEK-293T were cultured in the DMEM supplemented with 10% (v/v) fetal bovine serum (FBS) in a humidified atmosphere of 5% CO₂ at 37 °C. All treatments in hepatocytes are indicated in the figure legends.

**Transfection in hepatocytes**. For HADHA, BDH1 and HDAC7 knockdown, primary hepatocytes were transfected with siRNA using Lipofectamine® 2000 transfection reagent (ThermoFisher, America). Negative siRNA was used as a normal control. For HADHA, FOXO1 and HDAC7 overexpression, cells were transfected with plasmids according to the manufacturer's protocol. Empty vectors (pcDNA3.1) were transfected as control. Briefly, the culture medium was changed to Opti-MEM prior to transfection, and the mixture of 2.5 μL/ml Lipofectamine® 2000 and 1 μg/ml of each siRNA or plasmids pool were added to the cells. Media containing 10% FBS was added to the cells after 6 h incubation at 37 °C. The cells were harvested after 24 h post-transfection. HDAC7 mutation plasmids were bought from the Public Protein/Plasmid Library in China. The siRNA oligos used are as follows:

NC siRNA: UUCUCCGAACGUGUCACGUTT
ACGUGACACGUUCGGAGAATT;
HADHA siRNA: GGUUGCCAUUUCAUGCCAATT
UUGGCAUGAAAUGGCAACCTT;
BDH1siRNA: GCAGGAAGUACUUCGAUGAAA
UUUCAUCGAAGUACUUCCUGC;
HDAC7 siRNA: GCAGUGUGGUCAAGCAGAA
UUCUGCUUGACCACACUGC.

**Stable isotope tracing by liquid chromatography coupled with triple quadru-pole mass spectrometer (LC–MS/MS)**. For stable isotope tracing, the [U-¹³C] palmitate was conjugated with phosphate buffer saline (PBS) containing 6.02 mM bovine serum albumin (BSA). Primary hepatocytes were transfected with HADHA plasmid or siRNA for 24 h and treated with [U-¹³C]palmitate (0.1 mM) for 4 h. For the extraction of BHB and AcAc, primary hepatocytes were quickly washed once with ice-cold PBS and then sonicated with deionized water for 10 min. The crushed cell suspension was subsequently mixed with 200 μL of acetonitrile by vortexing for

2 min and centrifuging at 13,000 rpm (~12,000 × g) at 4 °C for 10 min. An aliquot of 80 μL of supernatants was obtained for further derivatization reaction. In brief, the 80 μL of supernatants were first transferred into a 1.5 mL tube, and then 40 μL of 3-nitrophenylhydrazine (150 mM) as the derivatization reagent and 40 μL of N-(3-dimethylaminopropyl)-N'-ethylcarbodiimide hydrochloride (240 mM) were added. The mixture was set at 30 °C and shaken for 40 min to complete the derivatization. After centrifugation (12,000 × g, 4 °C, 5 min), 2 μL of supernatant was injected for further liquid chromatography-tandem mass spectrometry analysis.

To extract Cit, primary hepatocytes were first washed once with ice-cold PBS to remove the medium components and subsequently rinsed with water. Then the cells were ultrasound-crushed in 1 mL of methanol/chloroform (9:1, v/v) solution for 10 min. After centrifugation (12,000 × g, 4 °C, 10 min), 900 μL of supernatants were carefully transferred and dried by a gentle stream of nitrogen gas. Finally, the residue was reconstituted with 100 μL of 50% acetonitrile aqueous solution, and 2 μL injection was analyzed by mass spectrometry.

Targeted detection was performed on a LC-20A system coupled to a triple quadrupole mass spectrometer (Shimadzu, LC-MS/MS 8050) operating in the negative ion mode. The chromatographic separation was achieved on a reversed-phase HSST3 column (Waters, 2.1 × 100 mm, 1.8 μm) maintained at 40 °C at a flow rate of 0.2 mL/min. The mobile phase consisted of water with 5 mM ammonium acetate (A) and acetonitrile (B). The gradient elution program was 5–20% B at 0–3 min, 20–75% B at 3–7 min, 75–95% B at 7–11 min, and then back to initial conditions, with 3 min for equilibration. The ESI source parameters were set as follows: DL temperature, 250 °C; interface temperature, 300 °C; heat block temperature, 400 °C; heating gas flow, 10 L/min; nebulizing gas flow, 3 L/min; and drying gas flow, 10 L/min. Multiple reaction monitoring mode was applied and the detailed ion transitions were: BHB, $m/z$ 238 → 194 (M), $m/z$ 240 → 194 (M + 2), $m/z$ 240 → 196 (M + 2), $m/z$ 242 → 196 (M + 4); AcAc, $m/z$ 371 → 218 (M), $m/z$ 373 → 220 (M + 2), $m/z$ 375 → 222 (M + 4); and Cit, $m/z$ 191 → 111 (M), $m/z$ 193 → 113 (M + 2). Multiple reaction monitoring chromatograms were provided in Supplementary Data 1.

**Quantitative real-time reverse transcription PCR (qRT-PCR)**. The total mRNA was isolated using TRIzol$^{TM}$ Reagent (Invitrogen, America) according to the manufacturer's instructions. cDNA was synthesized with 2 μg of RNA using a cDNA Synthesis kit (Roche, Switzerland) to synthesize cDNAs. qRT-PCR was performed on the Roche LightCycler 96 System using the Fast SYBR Green Master Mix (Roche, Switzerland). All the primer pairs used are listed in Supplementary Table 1.

**Western blotting**. Cell and tissue proteins were lysed with a radio- immunoprecipitation assay lysis buffer with 1% protease inhibitor cocktail. Nuclear extracts were prepared using an NE-PER Nuclear Cytoplasmic Extraction Reagent kit (Pierce, America). The total protein concentration was quantified with a BCA assay kit (Beyotime, China) for the normalization of assayed samples. Protein extracts were separated on 8–12% SDS-PAGE gels and transferred onto nitrocellulose membranes. The membranes were blocked by 5% nonfat milk and incubated with the indicated primary antibody (Supplementary Table 2) solution overnight at 4 °C followed by incubation with horse radish peroxidase-conjugated secondary antibodies.

**Immunofluorescence**. Primary hepatocytes were seeded in cell dish and fixed with 4% paraformaldehyde solution for 15 min, followed by permeabilization with 0.2% Triton X-100. Cells were then incubated in blocking buffer for 60 min and in the primary antibodies for overnight at 4 °C. After incubation with secondary antibodies for 60 min and DAPI for 10 min, the cells were analyzed using a confocal scanning microscope (Olympus, Japan).

**Immunoprecipitation**. Liver tissue and cell samples were harvested and lysed in 500 μL NP-40 lysate buffer. Lysates were centrifuged (12,000 × g, 10 min, 4 °C), and the supernatants were collected. After measuring the protein concentrations using the BCA assay kit (Beyotime, China), a 400 μg protein was transferred and incubated with 1 μL IgG or the primary antibodies on a rotator overnight at 4 °C. Then, 25 μL protein A/G magnetic agarose beads (MCE, China) were added and incubated with the lysates for another 6 h. Subsequently, the beads were washed with the PBS 8 times followed by the addition of 40 μL 1 × SDS-PAGE loading buffer and western blotting analysis.

For BHB-biotin pull down assays, cell lysates were preincubated with free biotin or biotin-labeled BHB (RuixiBiotech, China) for 4 h at room temperature, and then incubated with magic Dynabeads MyOne Streptavidin T1 (ThermoFisher, America) overnight at 4 °C on a rotator. The beads were washed 3–4 times and analyzed by western blotting.

**Luciferase reporter assay**. HEK-293 T cells were transfected with *Pck1* and *G6pc* luciferase reporter constructs plasmid (1 μg/mL). Renilla Luciferase plasmid (0.1 μg/mL) was used as an internal control. After 24 h transfection, cell lysates

were used for luciferase assays using a 96-well luminometer with a luciferase substrate system (Promega, America).

**Hepatic glucose production**. Primary hepatocytes were maintained in the DMEM with 10% FBS. Then the medium was replaced with Krebs-Ringer HEPES (KRH) buffer to fast the cells for 2 h. After washing with PBS two times, the cells were incubated in KRH buffer supplemented with 10 mM pyruvate, 100 nM glucagon for 6 h. The cell supernatant was then collected for glucose analysis using the Glucose Assay Kit (Jiancheng bioengineering, China), and normalized by total cellular protein content.

**Biochemical assays**. BHB contents were assayed with kits from commercial sources (Mlbio, China). The enzyme activity of HDAC4, 5 and 7 were determined with commercial kits (Abnova, Taiwan, China). Glucagon and insulin were determined with Elisa Kits from Cusabio and Jiancheng bioengineering, respectively.

**Molecular docking**. Coordinates of the crystal structure of HDAC7 were downloaded from the Protein Data Bank (PDB, https://www.rcsb.org/) with the ID 3c0z. The structure of BHB molecule was generated with the Chem3D tool in Chemoffice software. The AutoDock program (version 4.2) was employed to generate an ensemble of docked conformations for the BHB molecule bound to HDAC7. The objective docking method was utilized, in which a distinct large-sized cavity on the protein was chosen as the possible molecule binding area. We used the GA for conformational search, and all the C-C bonds of BHB were set to be rotatable. To explore the conformational space of the ligand as completely as possible, we performed 100 individual GA runs to generate 100 docked conformations. The size of the docking box was 60 Å × 60 Å × 60 Å with grid spacing of 0.375 Å. The docking box was centered at the mass center of the binding area and was large enough to enclose the whole binding area. The protein structure was kept fixed during molecular docking.

**SPR binding experiment**. The binding kinetics of BHB were assayed via SPR using a Biacore T200 instrument and manufacturer provided software (GE Healthcare, America). All measurements were performed at 25 °C. HDAC4 (4 μg/mL), HDAC5 (3 μg/mL) and HDAC7 (10 μg/mL) were diluted with 10 mM phosphate solution (pH 5.5, 4.5 and 4) and then covalently immobilized on CM5 chips. The proteins coupling solution was injected over the activated chip surface to achieve an immobilization level of 5,000 ~ 10,000 resonance units (RU). A blank surface was similarly treated but without any protein solution and used as a reference surface. Briefly, the running buffer during immobilization was HSB-P + Buffer (10 mM HEPES, 150 mM NaCl, 0.05% Tween20, pH 7.4) after activating the chip with 75 mg/mL N-(3-dimethylaminopropyl)-N'-ethylcarbodiimide hydrochloride and 10 mg/mL N-hydroxysuccinimide, and the coupling procedure was run at the flow rate of 5 μL/min. Binding affinity measurements were performed at flow rate of 30 μL/min using HSB-P + Buffer. BHB was assayed using single cycle kinetics mode as provided by the Biacore T200 control software. BHB was diluted in the running buffer and injected in a series of increasing concentrations for contact time of 450 s. Sensor grams were processed and analyzed using Biacore T200 evaluation software and the binding curves were fitted to determine the equilibrium dissociation constant (KD).

**Cellular thermal shift assay**. Primary hepatocytes were freeze-thawed 3 times using liquid nitrogen. The resultant cell lysate was incubated with BHB (50 μM) for 6 h at room temperature. Then, an aliquot of the lysate was sequentially heated at increasing temperature. After heating, the lysates were centrifuged, and HDAC7 abundance level in the supernatant was analyzed by immunoblotting.

**Statistics**. All results are given as mean ± standard error of the mean (SEM). Comparisons between two groups were analyzed by using a two-tailed Student's *t* test, and those among three or more groups by using one-way analysis of variance (ANOVA). Differences were considered significant at $p < 0.05$. Statistical significance analyses were performed using GraphPad Prism version 8.0.

**Reporting summary**. Further information on research design is available in the Nature Research Reporting Summary linked to this article.

## Data availability

The crystal structure of HDAC7 used in this study was downloaded from the Protein Data Bank with the PDB ID 3c0z (https://www.wwpdb.org/pdb?id=pdb_00003c0z). The data supporting this study are available in the Article and Supplementary Information. The uncropped gel images are included in the source data file. Source data are provided with this paper.

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

## Acknowledgements

This work was supported by the National Natural Science Fund of China for Distinguished Young Scholars (No. 81825023 to L.-W.Q.), Excellent Youth Foundation of Jiangsu Scientific Committee (BK20200082 to Q.L.) and National Natural Science Foundation of China (Nos. 82174036 and 81973550 to Q.L.).

## Author contributions

Q.L. and L.-W.Q. conceived and designed the experiments. A.P., X.-M.S., X.W., and F.-Q. H. performed the experiments. A.P., J.-F.L. and Y.-Y.C. analyzed the data. B.-L.L., Q.L., and A.P., wrote the manuscript. L.-W.Q., P.L. and R.-N.A. improved the manuscript. All authors contributed to the discussion of results and manuscript corrections.

## Competing interests

The authors declare no competing interests.

## Additional information

**Peer review information** *Nature Communications* thanks Cholsoon Jang, Gary Lopaschuk and the other anonymous reviewer(s) for their contribution to the peer review this work. Peer reviewer reports are available.

