## [Peer Review File · Nature Communications]

Reviewers' Comments:

Reviewer #1:

Remarks to the Author:

Pan et al. has conducted a study attempting to understand the regulatory role of ketone body beta-hydroxybutyrate (BHB) on hepatic glucagon signaling. Although the authors have done impressive amount of works to test their hypothesis, there are major concerns remained.

Major criticisms:

1) The authors found that HADHA knockdown rises blood glucose (figure 1). This was previously reported (PMID 26499439). This can be in part due to inhibition of fat oxidation, hepatic fat accumulation and consequent insulin resistance. On the other hand, the author's main thrust is that HADHA promotes BHB, which modulates hepatic glucagon response. To separate insulin effect, they performed primary hepatocyte experiments, but these experiments have other issues (see below) and do not fully explain the dramatic effect of HADAH modulation on glucose levels. Thus, data on insulin levels and hepatic insulin sensitivity are required for in vivo studies. Same goes for the BHB treatment in figure 3. This is particularly important because BHB increases insulin secretion (PMID 4913711, PMID 3307630). To prove the importance of BHB, they performed BDH1 knockdown experiments. However, in vivo (figure 6g), BDH1 knockdown reverses HADHA's drastic inhibitory effect on fat deposition (and likely insulin resistance). Given this BDH1 suppression effect not only on BHB production but also fat accumulation, BDH1 knockdown alone does not provide enough evidence for potential role of BHB. The authors may consider blocking insulin release (e.g., with somatostatin) or comparable treatment.

2) In figure 2, the method and interpretation of tracing experiments raises some concerns. Did the authors use free palmitate or conjugate it with albumin? Does media contain serum when they perform tracing? In the text, it says 6 hour incubation but in the method, it is 4 hour. Do the authors know this is steady-state labeling? Additionally, they only show relative abundance of labeled metabolites, so it is hard to judge biological importance. How much is citrate labeled from palmitate? 0.1% or 10%? In this regard, the authors should provide unlabeled metabolite fractions. They also mention in the abstract, "reduced acetyl-CoA" (line 30), but this reviewer cannot find the data.

3) While the authors argue that BHB is important for glucagon signaling, they did not provide blood BHB levels or BHB levels in hepatocytes with HADHA/BDH manipulations, all of which are critical information to support their claim. It is also unclear why the authors used 400 uM BHB for all their hepatocyte culture even though the Kd of BHB for HDAC binding or BHB serum or liver levels are an order of magnitude lower. The authors should perform a dose-response study to justify this concentration.

1) There are critical control groups missing. For figure 1, please include AAV-HADHA shRNA group for 1c and 1d. For figure 1f and g, AAV-HADHA group should be added. This is important because of the very strong HADHA effect on hepatic fat oxidation and accumulation, regardless of ketogenesis. Same for fig 2, which lacks HADHA plasmid alone. Supplementary figure 3 data should be combined with figure 2, within the same graphs. For fig 3a-c, BHB alone group is missing.

Minor issues

- 1) Fig.5b quality is poor. It is hard to agree the statement (line 213) that "BHB addition obviously reduced the binding" without quantitation data.
- 2) Figure 4b has a typo where BDH1 siRNA, fourth row, should be a plus instead of negative.
- 3) (line 256) "HADCs" should be "HDACs"

Reviewer #2:

Remarks to the Author:

The following issues should be addressed regarding the SPR experiment and the molecular docking:

- The quality of SPR profiles (as shown in Fig. 5c) seems poor. The RU amplitudes are small, and the curves at different ligand concentrations do not separate from either other in a reasonable

manner, rendering the result unreliable. Judging from these data, it is hard to rule out the possibility of non-specific binding. In addition, the authors should show the fitting curves, along with the parameters such as CHI, correlation coefficient, and the standard deviation.

- To obtain KD, the better choice is to measure the steady-state RU at different ligand concentrations.

- The docking part in Methods lacks a lot of details. The authors should provide a complete procedure that is detailed enough for other people to reproduce the result. This procedure should include (but not limited to) the PDB ID, blind docking (targeting the whole surface) or objective docking (targeting a specific region), parameters of the searching grid, which torsional angles of the ligand are rotatable, how to obtain pdb of the ligand, and so on.

- The docking scores and binding pockets of different poses should be described as well. The authors should also try docking BHB onto HDAC4 and HDAC5 (as long as their 3D structures are available).

- In general, docking is not regarded as a quantitatively reliable approach. The evidence of docking and mutagenesis is not enough for the authors to claim that BHB binds to Glu543. To claim this, a high-resolution structure of the complex (or an array of convincing indirect evidence) should be provided.

- Could the authors comment on why the binding of BHB inhibits enzyme activity of HDAC7 from the perspective of structure?

Other issues:

- In the main text (Line 237), "... but not Glu543 and..." should read "... but not His541 and...".

- In Methods (Line 508 and 510), "Kon" and "Koff" should be "kon" and "koff", respectively.

- In Fig. 5h, an enlarged inset showing interaction between BHB and HDAC7 should be provided as well.

Reviewer #3:

Remarks to the Author:

This study investigated the role of mitochondrial trifunctional protein subunit A (HADHA) and ketones in hepatic glucagon response. The authors found that glucagon inhibited ketogenesis due to reduced acetyl-CoA production by suppressing HADHA, a key enzyme in fatty acid oxidation. They found that the ketone body, β -hydroxybutyrate (BHB), inhibited Class IIa histone deacetylase 7 (HDAC7) activity via interaction with the residue of Glu543. HDAC7 inactivation by BHB preserved FoxO1 acetylation to facilitate FoxO1 nuclear exclusion, resulting in an inhibition of hepatic gluconeogenesis. High-fat-diet (HFD) feeding impaired ketogenesis in mice, while hepatic HADHA overexpression restored BHB production with improved metabolic parameters. The protective effects of HADHA were abrogated when hepatic 3-hydroxybutyrate dehydrogenase 1 was knocked down in HFD-fed mice.

This study provides a potentially novel explanation for how glucagon inhibits ketogenesis. However, I have a number of conceptual problems with how glucagon suppression of HADHA would decrease ketogenesis, as well as the lack of data to link mitochondrial HADHA suppression to modifications in nuclear FoxO1 expression and translocation. In addition, the relative importance of mitochondrial BHB versus nuclear BHB in preventing hepatic gluconeogenesis is confusing, and some of the data does not support a role for BHB acting at a mitochondrial level. These concerns are highlighted below:

Specific Comments:

1) The data showing that BHB may interact and inhibit HDAC7, thereby preserving FoxO1 acetylation is interesting, although not that novel, as BHB has already been shown to be an HDAC inhibitor. However, data is lacking to convincingly demonstrate that decreasing BHB decreases

FoxO1 acetylation, decreases phosphorylation and increases FoxO1 nuclear translocation.

2) Page 8, line 171: It is surprising that BDH1 knockdown (Supplementary Fig. 4a) blocked the HADHA overexpression-inhibited hepatic glucagon response in gluconeogenesis-associated genes and hepatic glucose production (Fig. 3e, f). How does this occur? Using AcAc, BDH1 increased expression of these genes. It suggests that BHB itself is important for expression of the gluconeogenic enzymes. Therefore, preventing BHB conversion to AcAc by knocking down BDH1 does not make sense.

3) BDH1 is a mitochondrial enzyme important in BHB oxidation. However, knocking down BDH1 decreased the expression of gluconeogenic enzymes. One would expect that knocking down BDH1 would increase BHB available for nuclear HDAC7 interaction, and therefore decrease gluconeogenic enzyme transcription. This possibility is not addressed.

4) It is not clear how downregulation of HADHA decrease ketogenesis. Normally, increases in fatty acid oxidation result in increased ketogenesis. This is opposite of what is expected, since HADHA is a key enzyme complex of fatty acid oxidation.

5) It is proposed that glucagon inhibition of HADHA decreases acetyl CoA production from fatty acid oxidation. Evidence to support this is weak. Furthermore, what is happening to this acetyl CoA is not clear. Is it being used for FoxO1 acetylation? Data to support this is not provided. Furthermore, it is not clear how the acetyl CoA would be getting out of the mitochondria to alter nuclear acetylation.

6) Figure 1g: It appears that overexpression of HADHA in liver increases G6Pase and PGC-1a, and marginally increased PEPCK in mice compared to controls. Why? This suggests that there is some increase in gluconeogenesis in HADHA overexpressing mice. This is not what would be expected if HDADA deficiency is increasing gluconeogenesis.

7) Page 7, line 147: How was ¹³C-palmitic acid added to hepatocytes? A concentration of 0.1 mM was used. Was it bound to albumin and what was the albumin concentration used?

8) Page 7: What happens to label incorporation in citrate from ¹³C-palmitate when HADHA is decreased?

9) Figure 2D: Glucagon did not increase ACAC [M+2] abundance. This is not consistent with glucagon decreasing fatty acid acetyl CoA going to ketones.

10) Figure 3g: Data on the effects of BDH1 knockdown on AcAc effects on hepatic glucose production are missing. Why wasn't this presented?

11) Figure 4: Where is HADHA expressed or overexpressed? Is it in mitochondria? How does mitochondrial HADHA inhibit nuclear translocation of FoxO1 (or nuclear transcription)?

12) Figure 5: HADHA overexpression decreases glucagon effects on nuclear HDAC7. How does this occur? There is no data on what HADHA overexpression does to nuclear HDAC7 in the absence of glucagon.

13) No data is provided as to what these procedures are actually doing to acetyl CoA levels or ketone levels.

14) Figure 6 b and c: Why are serum levels of BHB presented as relative levels? Why not absolute levels?

15) Figure 6d: When liver BDH1 was silenced, glucose tolerance was worsened.

Minor:

Page 6, line 130: fasting blood "glucose"...

Reviewer #4:

Remarks to the Author:

In this manuscript Pan, et al. investigate novel mechanisms underlying hepatic gluconeogenesis (GNG). The primary focus of this work is on the dysregulation of HADHA in response to glucagon or hyperglucagonemia observed after HF-feeding. The authors further implicate the HADHA metabolite, BHB, as critical inhibitor of HDAC7. Subsequently, acetylation/cytosolic retention of FoxO1 is lost and GNG is unrestricted. The data set is intriguing and expansive in scope. The authors should be commended on the comprehensive mechanistic studies. However, the article is poorly written with little acknowledgment of prior studies, nor explanation as to why their results are in complete opposition (especially in regards to glucagon and BHB dynamics). While the experiments are expansive, details as to how they were conducted are lacking. Likewise, controls

for many studies are missing, making interpretation of the results impossible. Together these issues (detailed below) dramatically diminish my enthusiasm for this work.

Major:

1. The primary weakness of this report is a superficial nod to past studies and a lack of discussion how such vastly different effects are possible.
 - a. Pharmacological activation of the glucagon receptor elevates BHB, yet no mention of this confounding data is made.
 - b. Glucagon secretion is more tightly coupled to amino acid and fatty acid exposure (post prandial) than to hypoglycemia. This makes the current study difficult to reconcile with the physiology.
2. Experimental conditions are brief or lacking (i.e. doses, time of treatment, prandial state of the mice at tissue collection)
3. Critical controls are routinely omitted. Several examples are included below, but this is not fully inclusive of all missing control groups.
 - a. Fig 1c,d,f,g – no vehicle treated AAV-HADHA group
 - b. Fig 2b-e – same
 - c. Fig 3a-d – no BHB (mono-treatment) control
4. Why is FOXO1 overexpressed in Fig5? No control for this variable nor rationale is included.
5. Why were HepG2 cells used for only 2 experiments (fig 5)?
6. Confirmation of overexpression and/or knockdown (KD) should be included for all targets, but especially for those of studies detailed in Fig 6, where genetic manipulation is conducted prior to 12w of HF-feeding. mRNA and Protein levels should be confirmed at the time of experiment. Current data included in the supplemental data is unclear as to timing and KD is partial in most targets, this needs to be addressed.

Minor:

1. The introduction (and really the article in general) is poorly constructed. As written, it reads like a loosely associated collection of statements.
2. Gene/protein nomenclature is muddled. The authors should review the current mouse standards and revise.

Summary of point-by-point response to reviewers

We thank the four reviewers for their comprehensive comments on our initial submission (NCOMMS-21-06349-T) and their highly constructive suggestions. A detailed point-by-point response to all the reviewer's comments are provided in this document. We performed a series of additional experiments to address the reviewers' comments, and generated new data to that effect. We believe that these additional experiments bolster the manuscript's central claims and have significantly strengthened our conclusion. We have thoroughly reorganized and rewritten the text to make it easier for the readers to follow. A native English speaker, Dr. Raphael N. Alolga, was invited to help edit our manuscript.

REVIEWER COMMENTS

Reviewer #1 (Remarks to the Author):

Pan et al. has conducted a study attempting to understand the regulatory role of ketone body beta-hydroxybutyrate (BHB) on hepatic glucagon signaling. Although the authors have done impressive amount of works to test their hypothesis, there are major concerns remained.

Response: We thank the Reviewer for constructive comments on our manuscript.

Major criticisms:

- 1) The authors found that HADHA knockdown rises blood glucose (figure 1). This was previously reported (PMID 26499439). This can be in part due to inhibition of fat oxidation, hepatic fat accumulation and consequent insulin resistance. On the other hand, the author's main thrust is that HADHA promotes BHB, which modulates hepatic glucagon response. To separate insulin effect, they performed primary hepatocyte experiments, but these experiments have other issues (see below) and do not fully explain the dramatic effect of HADAH modulation on glucose levels. Thus, data on

insulin levels and hepatic insulin sensitivity are required for in vivo studies. Same goes for the BHB treatment in figure 3. This is particularly important because BHB increases insulin secretion (PMID 4913711, PMID 3307630). To prove the importance of BHB, they performed BDH1 knockdown experiments. However, in vivo (figure 6g), BDH1 knockdown reverses HADHA's drastic inhibitory effect on fat deposition (and likely insulin resistance). Given this BDH1 suppression effect not only on BHB production but also fat accumulation, BDH1 knockdown alone does not provide enough evidence for potential role of BHB. The authors may consider blocking insulin release (e.g., with somatostatin) or comparable treatment.

Response: We thank the reviewer for this suggestion. We have performed all the suggested experiments so as to lend more credence to our conclusions. Glucagon is reported to stimulate insulin secretion through the GLP-1 and glucagon receptors on pancreatic β cells (JCI Insight, 2019, 4: e129954). As expected, acute glucagon challenge exhibited the potential to promote to promote insulin secretion in mice, and HADHA overexpression significantly increased insulin secretion (Supplementary Fig. 4a). To exclude potential effect of insulin, we blocked insulin release with somatostatin (Supplementary Fig. 4a). Somatostatin did not alter the inhibitory effects of HADHA overexpression on blood glucose in glucagon-challenged mice (Supplementary Fig. 4b). In close agreement, somatostatin did not reverse the inhibitory effect of HADHA overexpression on gene expressions of *Pck1*, *G6pc* and *Pgc1a* in the liver of mice (Supplementary Fig. 4c). These results suggested that HADHA suppressed gluconeogenesis independently of insulin. We have added the relevant information (Lines 143-146), figures (Supplementary Fig. 4a-c) and reference (ref. 22) in the Revised Manuscript.

As the reviewer mentioned, ketone bodies were reported to induce insulin secretion (Nature, 1970, 227: 384-385; Archives of Biochemistry Biophysics, 1987, 257: 140-143). Our results showed that BHB moderately promoted insulin secretion in mice (Supplementary Fig. 7a). To separate insulin effect, we treated mice with somatostatin to block insulin release (Supplementary Fig. 7a). Somatostatin showed slight but no significant effects on BHB inhibition of blood glucose in glucagon-challenged mice

(Supplementary Fig. 7b). More importantly, the inhibitory effects of BHB on hepatic *Pck1*, *G6pc* and *Pgc1a* expressions were not altered by somatostatin (Supplementary Fig. 7c). These results indicated that BHB inhibited gluconeogenesis independently of insulin. We have added the relevant information (Lines 181-184), figures (Supplementary Fig. 7a-c) and references (ref. 23-24) in the Revised Manuscript.

Supplementary Fig. 4 HADHA inhibited hepatic glucagon response independently of insulin release. **a** Serum insulin levels of the mice treated with AAV8-HADHA or somatostatin (3 mg/kg, 15 min), glucagon stimulation for 1 h (n = 5). **b** Blood glucose levels in normal mice subjected to glucagon challenge (2 mg/kg, 1 h) and treated with AAV8-HADHA or somatostatin (3 mg/kg, 15 min). AUC is indicated on the right (n = 5). **c** mRNA levels of *Pck1*, *G6pc* and *Pgc1a* in the livers of the mice in panel **b** (n = 5). AUC area under the curve, GLC glucagon, NC normal control, ns no significance. Bars represent mean \pm SEM values. Statistical differences were determined by one-way ANOVA. $**p < 0.01$ vs. the control group. Source data are provided as a Source Data file.

Supplementary Fig. 7 BHB inhibited glucagon response independently of insulin release. **a** Serum insulin levels in the mice subjected to glucagon challenge (2 mg/kg), treatment with BHB (100 mg/kg) and somatostatin (3 mg/kg, 15 min). **b** Blood glucose levels in normal mice subjected to glucagon challenge (2 mg/kg), treatment with BHB (100 mg/kg) and somatostatin (3 mg/kg, 15 min). AUC is indicated on the right (n = 5). **c** mRNA levels of *Pck1*, *G6pc*, and *Pgc1a* in the livers of the mice in panel **b** (n = 5). AUC area under the curve, BHB β -hydroxybutyrate, GLC glucagon, NC normal control, ns no significance. Bars represent mean \pm SEM values. Statistical differences were determined by one-way ANOVA. * $p < 0.05$ vs. the control group, ** $p < 0.01$ vs. the control group. Source data are provided as a Source Data file.

2) In figure 2, the method and interpretation of tracing experiments raises some concerns. Did the authors use free palmitate or conjugate it with albumin? Does media contain serum when they perform tracing? In the text, it says 6 hours incubation but in the method, it is 4 hour. Do the authors know this is steady-state labeling? Additionally, they only show relative abundance of labeled metabolites, so it is hard to judge biological importance. How much is citrate labeled from palmitate? 0.1% or 10%? In this regard, the authors should provide unlabeled metabolite fractions. They also mention in the abstract, “reduced acetyl-CoA” (line 30), but this reviewer cannot find the data.

Response: We apologize for the lack of clarity in the experimental methods. We used bovine serum albumin (BSA) at a concentration of 40% with the reagent buffer to conjugate [U-¹³C]palmitic acid (20 mM). A final concentration of 0.1 mM conjugated

[U-¹³C]palmitic acid was added to the primary hepatocytes and incubated for 4 h incubation. When performing tracing, we used media containing 2% serum. We have effected these corrections in the Methods (Lines 453-457) and Figure Legends (Lines 725-728) of the Revised Manuscript.

[U-¹³C]palmitic acid employed in this work is a steady-state labeling. Per the reviewer's suggestion, we have provided information on the unlabeled metabolite fractions (M0) for BHB, AcAc and Cit (Fig. 2b-d). Both the labeled (M2 and M4) and unlabeled (M0) metabolite fractions analyses showed that glucagon significantly inhibited the production of BHB (Fig. 2b) and AcAc (Fig. 2c). In addition, glucagon stimulation inhibited the metabolite Cit, a TCA cycle intermediate (Fig. 2d). HADHA overexpression promoted Cit in the absence or presence of glucagon stimulation (Fig. 2d). The amount of labeled Cit as Cit (M2) from [U-¹³C]palmitic acid (0.1 mM) accounts for approximately 20% of the total (M0+M2) (Fig. 2d). HADHA overexpression considerably enhanced BHB and AcAc generation, and restored ketone body production that was inhibited by glucagon (Fig. 2b, c). In contrast, HADHA downregulation inhibited production of BHB, AcAc and Cit, and further aggravated the inhibitory effects of glucagon in ketogenesis (Fig. 2e-g). We have correspondingly added the relevant information (Lines 157-167) and figures (Fig. 2a-g) in the Revised Manuscript.

We performed additional experiments to determine the levels of acetyl-CoA in hepatocytes. Results showed that HADHA overexpression increased the levels of acetyl-CoA (Supplementary Fig. 5). In contrast, glucagon stimulation decreased the amount of acetyl-CoA, and this effect could be reversed by HADHA overexpression. Related information (Lines 29-30 in Abstract and Lines 150-153 in Results) and figures (Supplementary Fig. 5) has been added in the Revised Manuscript.

Fig. 2 Stable isotope tracing of ketogenesis. **a** A schematic summary of isotope tracing of ketogenesis using [U-¹³C]palmitate. **(b-d)** Primary hepatocytes treated with [U-¹³C]palmitate (0.1 mM) for 4 h following 100 nM glucagon stimulation for 1 h with or without HADHA plasmid transfection (n = 3-8). Acetyl-CoA serves as a precursor of ketogenesis or TCA cycle. **b** BHB relative abundance. **c** AcAc relative abundance. **d** Cit relative abundance. **(e-g)** Primary hepatocytes treated with [U-¹³C]palmitate (0.1 mM) for 4 h following 100 nM glucagon stimulation for 1 h with or without HADHA siRNA transfection (n = 3-5). **e** BHB relative abundance. **f** AcAc relative abundance. **g** Cit relative abundance. AcAc acetoacetate, BHB β-hydroxybutyrate, Cit citrate, GLC glucagon, TCA tricarboxylic acid cycle. Bars represent mean ± SEM values. Statistical differences were determined by one-way ANOVA. **p* < 0.05, ***p* < 0.01 vs. the control group; #*p* < 0.05, ##*p* < 0.01 vs. indicated treatments. Source data are provided as a Source Data file.

Supplementary Fig. 5 HADHA promoted acetyl-CoA production. Acetyl-CoA levels in HADHA-overexpressed primary hepatocytes with or without 100 nM glucagon stimulation for 1 h (n = 5). GLC glucagon. Bars represent mean ± SEM values. Statistical differences were determined by one-way ANOVA. ***p* < 0.01 vs. the control group, ##*p* < 0.01 vs. indicated treatments. Source data are provided as a Source Data file.

3) While the authors argue that BHB is important for glucagon signaling, they did not provide blood BHB levels or BHB levels in hepatocytes with HADHA/BDH

manipulations, all of which are critical information to support their claim. It is also unclear why the authors used 400 μ M BHB for all their hepatocyte culture even though the Kd of BHB for HDAC binding or BHB serum or liver levels are an order of magnitude lower. The authors should perform a dose-response study to justify this concentration.

Response: To address the reviewer's comments, we tested the levels of BHB with HADHA or BDH1 manipulations in vitro and in vivo. In primary hepatocytes, HADHA overexpression restored glucagon-inhibited BHB level, but this effect was diminished by BDH1 knockdown (Supplementary Fig. 10b). In mice, HADHA overexpression increased serum BHB levels with or without glucagon challenge (Supplementary Fig. 6b). In contrast, liver-specific HADHA knockdown decreased serum BHB levels in the absence of glucagon and further augmented BHB decline in the presence of glucagon (Supplementary Fig. 6c). We have added the relevant information (Lines 170-173 and 191-193) and figures (Supplementary Fig. 6b, c and Supplementary Fig. 10b) in the Revised Manuscript.

To justify the concentration of BHB treatment, we performed a concentration-response study in primary hepatocytes using 50 μ M, 100 μ M, 200 μ M, 400 μ M, and 800 μ M. Results showed that BHB inhibited glucagon-stimulated gluconeogenesis-associated genes (mRNA abundance of *Pck1*, *G6pc* and *Pgc1a*) in a dose-dependent manner (Supplementary Fig. 8). BHB at 400 μ M and 800 μ M displayed the most effective inhibitory effects on gluconeogenesis. We therefore chose the relatively lower concentration of 400 μ M exogenous BHB for the primary hepatocytes treatment. We agree with the reviewer that 400 μ M BHB used in this work is higher than the KD of BHB for HDAC binding or the levels of BHB in the serum (100-200 μ M) or the liver. To highlight the effects of signaling metabolites such as BHB, a higher concentration is usually used. The BHB concentration used in this work is similar to a published study, which used BHB at 500 μ M for cardiomyocytes (Circulation Research, 2021, 128(2): 232-245). We have added the relevant information (Lines 187-189) and figure (Supplementary Fig. 8) in the Revised Manuscript.

Supplementary Fig. 10b BHB level in HADHA-overexpressed primary hepatocytes transfected with BDH1 siRNA and stimulated with 100 nM glucagon for 1 h (n = 6). BDH1 β -hydroxybutyrate dehydrogenase 1, BHB β -hydroxybutyrate, GLC glucagon. Bars represent mean \pm SEM values. Statistical differences were determined by one-way ANOVA. ** $p < 0.01$ vs. the control group. ## $p < 0.01$ vs. indicated treatments. Source data are provided as a Source Data file.

Supplementary Fig. 6 (b, c) Serum BHB levels in HADHA liver-specific overexpression (b) or knockdown (c) mice with or without glucagon stimulation (2 mg/kg, n = 5). AAV adeno-associated virus, BHB β -hydroxybutyrate, GLC glucagon, NC normal control. Values represent mean \pm SEM. Statistical differences were determined by one-way ANOVA. * $p < 0.05$, ** $p < 0.01$ vs. the control group. ## $p < 0.01$ vs. indicated treatments. Source data are provided as a Source Data file.

Supplementary Fig. 8 BHB inhibited gluconeogenesis in a dose-dependent manner. a Relative mRNA abundance of *Pck1*, *G6pc* and *Pgc1a* in glucagon-stimulated (100

nM, 1 h) primary hepatocytes with or without BHB treatment (50, 100, 200, 400, and 800 μ M, n = 6). BHB β -hydroxybutyrate, GLC glucagon, PBS phosphate buffer solution. Bars represent mean \pm SEM values. Statistical differences were determined by one-way ANOVA. * p < 0.05. ** p < 0.01 vs. the control group. Source data are provided as a Source Data file.

4) There are critical control groups missing. For figure 1, please include AAV-HADHA shRNA group for 1c and 1d. For figure 1f and g, AAV-Hp ADHA group should be added. This is important because of the very strong HADHA effect on hepatic fat oxidation and accumulation, regardless of ketogenesis. Same for fig 2, which lacks HADHA plasmid alone. Supplementary figure 3 data should be combined with figure 2, within the same graphs. For fig 3a-c, BHB alone group is missing.

Response: To address the reviewer's comments, we performed all experiments with the inclusion of control groups. For Fig. 1, we included the AAV-HADHA shRNA group (1c and 1d). Results showed that hepatic HADHA knockdown alone increased fasting blood glucose levels and expression of gluconeogenesis-associated genes in mice (Fig. 1c, d). Upon glucagon stimulation, HADHA knockdown further aggravated blood glucose levels and gluconeogenic genes expression (Fig. 1c, d).

For Fig. 1f and g, AAV-HADHA group was added. Results showed that liver-specific HADHA overexpression alone slightly decreased fasting blood glucose and expression of gluconeogenesis-associated genes in mice (Fig. 1f, g). Upon glucagon stimulation, HADHA overexpression considerably decreased the glucagon-induced blood glucose levels and gluconeogenesis-associated genes expression (Fig. 1f, g).

For Fig. 2, we added the HADHA plasmid alone control, and combined initial Supplementary Fig. 3 data with Fig. 2. Both stable isotope tracing and unlabeled metabolite fractions showed that HADHA overexpression induced production of downstream metabolites such as BHB, AcAc and Cit in primary hepatocytes (Fig. 2b-d).

For Fig 3a, b, BHB alone control was added. Results showed that exogenous BHB alone is sufficient to decrease the mRNA levels of *G6pc* and *Pgc1a* (Fig. 3a, b). Upon glucagon challenge, BHB greatly inhibited blood glucose and gluconeogenesis-

associated genes expression.

We have added the relevant information (Lines 127-134, 157-167 and 179-181) and figures (Fig. 1b-g, Fig. 2b-d and Fig. 3a, b) in the Revised Manuscript.

Fig. 1 HADHA inhibited glucagon-stimulated hepatic gluconeogenesis. **a** Hepatic HADHA protein level in glucagon-challenged mice (2 mg/kg, n = 3). **b** Fasting blood glucose of liver-specific HADHA knockdown mice (n = 5). **c** Blood glucose levels in normal mice subjected to glucagon challenge (2 mg/kg) and treated with AAV8-HADHA shRNA or AAV8-NC. AUC is indicated on the right (n = 5). **d** mRNA levels of *Pck1*, *G6pc* and *Pgc1a* in the livers of the mice in panel **c** (n = 5). **e** Fasting blood glucose of liver-specific HADHA-overexpressed mice (n = 5). **f** Blood glucose curve and AUC for mice treated with AAV8-HADHA or AAV8-NC after glucagon injection (2 mg/kg, n = 5). **g** mRNA levels of *Pck1*, *G6pc* and *Pgc1a* in the livers of the mice in panel **f**. AAV adeno-associated virus, AUC area under the curve, GLC glucagon, NC normal control. Bars represent mean \pm SEM values. Statistical difference between two groups were determined by a two-tailed Student's *t* test, and all others were used one-way ANOVA. **p* < 0.05, ***p* < 0.01 vs. the control group. #*p* < 0.05, ###*p* < 0.01 vs. indicated treatments. Source data are provided as a Source Data file.

Fig. 2 (b-d) Primary hepatocytes treated with [U-¹³C]palmitate (0.1 mM) for 4 h following 100 nM glucagon stimulation for 1 h with or without HADHA plasmid transfection (n = 3-8). Acetyl-CoA serves as a precursor of ketogenesis or TCA cycle. **b** BHB relative abundance. **c** AcAc relative abundance. **d** Cit relative abundance. AcAc acetoacetate, BHB β -hydroxybutyrate, Cit citrate, GLC glucagon. Bars represent mean \pm SEM values. Statistical differences were determined by one-way ANOVA. **p* < 0.05, ***p* < 0.01 vs. the control group; #*p* < 0.05, ###*p* < 0.01 vs. indicated treatments. Source data are provided as a Source Data file.

Fig. 3 (a, b) **a** Blood glucose levels in normal mice subjected to glucagon challenge (2 mg/kg) and BHB treatment (100 mg/kg). AUC is indicated on the right (n = 5). **b** mRNA levels of *Pck1*, *G6pc* and *Pgc1a* in the livers of the mice in panel **a** (n = 5). AUC area under the curve, BHB β-hydroxybutyrate, GLC glucagon. Values represent mean ± SEM. Statistical differences were determined by one-way ANOVA. **p* < 0.05, ***p* < 0.01 vs. the control group. ###*p* < 0.01 vs. indicated treatments. Source data are provided as a Source Data file.

Minor issues

1) Fig.5b quality is poor. It is hard to agree the statement (line 213) that “BHB addition obviously reduced the binding” without quantitation data.

Response: We have repeated the experiment and the results are shown below (Fig. X1). It is evident for this figure that, HADHA overexpression obviously reduced the binding of HDAC7 to FOXO1, but showed no significant impact in respect of binding of HDAC4 and 5 to same. We have replaced the previous figure with one of high quality (Fig. 5b) in the Revised Manuscript.

Fig. X1 Immunoprecipitation analysis of interaction between FOXO1 and HDAC4, HDAC5 or HDAC7. Assay was tested in HADHA-overexpressed hepatocytes with or without BDH1 siRNA transfection and 100 nM glucagon stimulation for 1 h (n = 3). GLC glucagon, NC normal control.

2) Figure 4b has a typo where BDH1 siRNA, fourth row, should be a plus instead of

negative.

Response: We apologize for the mistake. We have corrected it in the Revised Manuscript.

3) (line 256) “HADCs” should be “HDACs”

Response: Sorry for the typographical error. This has been rectified in the Revised Manuscript.

Reviewer #2 (Remarks to the Author):

The following issues should be addressed regarding the SPR experiment and the molecular docking:

- The quality of SPR profiles (as shown in Fig. 5c) seems poor. The RU amplitudes are small, and the curves at different ligand concentrations do not separate from either other in a reasonable manner, rendering the result unreliable. Judging from these data, it is hard to rule out the possibility of non-specific binding. In addition, the authors should show the fitting curves, along with the parameters such as CHI, correlation coefficient, and the standard deviation.

Response: Based on the reviewer's comments, we modified our SPR protocol. The RU amplitudes were enlarged with ligand concentrations ranging from 0.39-6.25 μM to 0.045-12.35 μM to get a more qualified KD value. In addition, we have provided the fitting curves, along with the parameters such as Chi^2 , correlation coefficient, and the standard deviation (SD). Results showed that BHB exhibited favorable binding affinity to HDAC7 with $\text{KD} = 0.02 \mu\text{M}$, $\text{Chi}^2 = 0.142$, $R = 0.98$, and $\text{SD} = 4.00\text{E-}08$ (Fig. 6a). BHB displayed much weaker binding affinity to HDAC4 with $\text{KD} = 4.47 \mu\text{M}$, $\text{Chi}^2 = 0.0493$, $R = 0.95$, and $\text{SD} = 9.70\text{E-}07$, and HDAC5 with $\text{KD} = 3.19 \mu\text{M}$, $\text{Chi}^2 = 0.0804$, $R = 0.93$, and $\text{SD} = 6.00\text{E-}07$. We have added the relevant information (Lines 256-258) and figure (Fig. 6a) in the Revised Manuscript.

Fig. 6a The interaction of BHB with HDACs by SPR. a The interaction of BHB with HDACs by surface plasmon resonance. A series of BHB concentrations was injected onto the HDAC4, HDAC5 or HDAC7 biosensor surface. The frequency response, equilibrium dissociation constant (KD) and fitting curves were displayed. BHB β -hydroxybutyrate, SPR surface plasmon resonance.

- To obtain KD, the better choice is to measure the steady-state RU at different ligand concentrations.

Response: Thank you for the good suggestion. We have modified our experimental conditions accordingly. The RU amplitudes were enlarged with ligand concentrations ranging from 0.39-6.25 μM to 0.045-12.35 μM to get a more qualified KD value. Results showed that BHB exhibited favorable binding affinity to HDAC7 with KD = 0.02 μM , much higher than HDAC4 with KD = 4.47 μM and HDAC5 with KD = 3.19 μM . We have added the relevant information (Lines 256-258) and figure (Fig. 6a) in the Revised Manuscript.

Fig. 6a The interaction of BHB with HDACs by SPR. a The interaction of BHB with HDACs by surface plasmon resonance. A series of BHB concentrations was injected

onto the HDAC4, HDAC5 or HDAC7 biosensor surface. The frequency response, equilibrium dissociation constant (KD) and fitting curves were displayed. BHB β -hydroxybutyrate, SPR surface plasmon resonance.

- The docking part in Methods lacks a lot of details. The authors should provide a complete procedure that is detailed enough for other people to reproduce the result. This procedure should include (but not limited to) the PDB ID, blinding docking (targeting the whole surface) or objective docking (targeting a specific region), parameters of the searching grid, which torsional angles of the ligand are rotatable, how to obtain pdb of the ligand, and so on.

Response: We have provided a detailed docking methodological procedure that could easily replicated by other researchers in the Revised Manuscript (Lines 549-562). Coordinates of the crystal structure of HDAC7 was downloaded from the Protein Data Bank (PDB, <https://www.rcsb.org/>) with the PDB ID 3c0z. The structure of BHB molecule was generated with the Chem3D tool in Chemoffice software. The Autodock program (version 4.2) was employed to generate an ensemble of docked conformations for the BHB molecule bound to HDAC7. The objective docking method was utilized, in which a distinct large-sized cavity on the protein was chosen as the possible molecule binding area. We used the genetic algorithm (GA) for conformational search, and all the C-C bonds of BHB were set to be rotatable. To explore the conformational space of the ligand as completely as possible, we performed 100 individual GA runs to generate 100 docked conformations. The size of the docking box was $60 \text{ \AA} \times 60 \text{ \AA} \times 60 \text{ \AA}$ with grid spacing of 0.375 \AA . The docking box was centered at the mass center of the binding area, and is large enough to enclose the whole binding area. The protein structure was kept fixed during molecular docking.

- The docking scores and binding pockets of different poses should be described as well. The authors should also try docking BHB onto HDAC4 and HDAC5 (as long as their 3D structures are available).

Response: We apologize for the lack of adequate details. The top 5 binding poses for BHB to HDAC7 resided exactly at the same pocket, which involved at least three

hydrogen-bonds with His541, Glu543 and Asp626 (Fig. 6d). The docking scores for these top 5 binding poses were close to each other (-3.55 ~ -3.66 kcal/mol), as well as the root mean square deviations between their configurations ($< 0.2 \text{ \AA}$). Other binding poses deviated significantly from the top 5, and showed less favorable docking scores. We have added the relevant information (Lines 266-272) and figures (Fig. 6d) in the Revised Manuscript.

As the crystal structure of HDAC4, but not HDAC5, is available in the protein data bank, we performed the molecular docking of BHB onto HDAC4. A representative binding mode for BHB onto HDAC4 is shown in Fig. X2.

Fig. X2. The AutoDock predicted binding modes and sites of BHB with HDAC4. The protein is shown as cartoon and BHB as sticks. Arg arginase, BHB β -hydroxybutyrate, His histidine.

- In general, docking is not regarded as a quantitatively reliable approach. The evidence of docking and mutagenesis is not enough for the authors to claim that BHB binds to Glu543. To claim this, a high-resolution structure of the complex (or an array of convincing indirect evidence) should be provided.

Response: The concern of question the reviewer is very legitimate and very much appreciated. In an effort to specifically address this point, we added an array of evidential experiments. We employed an orthogonal target validation approach (CETSA) to confirm the BHB-HDAC7 interaction. CETSA recognizes ligand-target engagement by immunoblotting detection of increased stability of the target proteins towards heat-induced precipitation. We noted that BHB incubation stabilized HDAC7

at multiple assayed temperatures in the examined heat-denatured hepatocytes (Fig. 6b). Furthermore, we designed a BHB-biotin probe to investigate its interaction with HDAC7 by immunoblotting analysis. Results showed BHB-biotin probe effectively pulled HDAC7 protein out, and their binding was obviously reduced by addition of unlabeled BHB due to competitive binding (Fig. 6c).

AutoDock analysis showed that BHB formed three hydrogen-bonds with HDAC7 by interacting with His541, Glu543 and Asp626 (Fig. 6d). To examine the binding effect among these three sites, alanine residues were introduced by site-directed mutagenesis to replace each site. Immunoblotting assay showed that Glu543 mutation, but not His541 mutation or Asp626 mutation, impaired the binding of BHB-biotin to HDAC7 (Fig. 6e). Mutagenesis with alanine residue did not alter the HDAC7 enzyme activity (Fig. 6f). Compared with wild-type HDAC7, the inhibitory effects of BHB on *Pck1*, *G6pc* and *Pgc1a* were abolished by Glu543 mutation, but not by His541 or Asp626 mutation (Fig. 6g). Collectively, these data suggest that BHB possibly binds to HDAC7 Glu543 to inhibit glucagon response. We have added the relevant information (Lines 256-279) and figures (Fig. 6b-g) in the Revised Manuscript.

Fig. 6 (b-g) **b** CETSAs for in-cell HDAC7 target engagement. Representative western blots showed thermostable HDAC7 with indicated heat shocks in the presence or absence of BHB (50 μM, n = 3). **c** Immunoprecipitation analysis of interaction between biotin-BHB and HDAC7 in primary hepatocytes transfected with HDAC7 plasmid with or without unlabeled BHB treatment (n = 3). **d** Binding modes and sites of BHB with HDAC7 predicted by AutoDock. The protein is shown as cartoon and BHB as sticks. **e** Immunoprecipitation analysis of interaction between BHB-biotin and HDAC7 in 293T cells transfected with HDAC7, HDAC7 (Mut-His541), HDAC7 (Mut-Glu543) or HDAC7 (Mut-Asp626) plasmid (n = 3). **f** Enzyme activity of HDAC7 in HepG2 cells transfected with HDAC7 (WT), HDAC7 (Mut-His541), HDAC7 (Mut-Glu543) or

HDAC7 (Mut-Asp626) plasmid (n = 5). **g** Relative mRNA abundance of *Pck1*, *G6pc* and *Pgc1a* in primary hepatocytes transfected with HDAC7 (WT), HDAC7 (Mut-His541), HDAC7 (Mut-Glu543) or HDAC7 (Mut-Asp626) plasmid after BHB treatment (n = 6). Asp aspartic acid, BHB β -hydroxybutyrate, CETSA, cellular thermal shift assay, GLC glucagon, ns no significance, Glu glutamic acid, His histidine, Mut mutant, WT wild type. Values represent mean \pm SEM. Statistical differences were determined by one-way ANOVA. * $p < 0.05$, ** $p < 0.01$ vs. the control group. Source data are provided as a Source Data file.

- Could the authors comment on why the binding of BHB inhibits enzyme activity of HDAC7 from the perspective of structure?

Response: We provided an explanation for added some discussion to explain the inhibitory effects of BHB on HDAC7 enzyme activity from the perspective of its structure (Lines 351-356) as follows: “The binding cavity of HDAC7 is delimited by the amino acids His531, His541, Pro542, Glu543, Ile628 and Phe679 (Journal of Biological Chemistry, 2008, 283: 11355-11363). The docked pose of BHB, structurally similar to the canonical HDAC inhibitor butyrate, shows interactions with Glu543 (Bioorganic and Medicinal Chemistry, 2013, 21: 3795-37807). Thus, the BHB molecule, which is able to establish contacts with this unique pocket, can be a structural element addressing inhibition for this enzyme.”

Other issues:

- In the main text (Line 237), “... but not Glu543 and...” should reads ““... but not His541 and...””.

Response: We have corrected it in the Revised Manuscript.

- In Methods (Line 508 and 510), “Kon” and “Koff” should be “kon” and “koff”, respectively.

Response: These mistakes have been corrected in the Revised Manuscript.

- In Fig. 5h, an enlarged inset showing interaction between BHB and HDAC7 should be provided as well.

Response: We thank the reviewer for this advice. We have added an enlarged inset

showing interaction between BHB and HDAC7. Please see the Fig. 6d in the Revised Manuscript.

Fig. 6d Binding modes and sites of BHB with HDAC7 predicted by AutoDock. The protein is shown as cartoon and BHB as sticks.

Reviewer #3 (Remarks to the Author):

This study provides a potentially novel explanation for how glucagon inhibits ketogenesis. However, I have a number of conceptual problems with how glucagon suppression of HADHA would decrease ketogenesis, as well as the lack of data to link mitochondrial HADHA suppression to modifications in nuclear FoxO1 expression and translocation. In addition, the relative importance of mitochondrial BHB versus nuclear BHB in preventing hepatic gluconeogenesis is confusing, and some of the data does not support a role for BHB acting at a mitochondrial level. These concerns are highlighted below:

Response: We thank the reviewer for the positive comments and the concerns raised about our manuscript. We have performed a series of extensive experiments to address the reviewer's comments and concerns. Importantly, we believe that these additional experiments bolster the manuscript's central claims and have significantly strengthened the novelty of our work. We provide below a point-by-point response to specific comments by the reviewer.

Specific Comments:

1) The data showing that BHB may interact and inhibit HDAC7, thereby preserving FoxO1 acetylation is interesting, although not that novel, as BHB has already been shown to be an HDAC inhibitor. However, data is lacking to convincingly demonstrate that decreasing BHB decreases FoxO1 acetylation, decreases phosphorylation and increases FoxO1 nuclear translocation.

Response: To address the reviewer's concern, we have performed a series of experiments to confirm the effect of BHB on FOXO1 acetylation, phosphorylation and nuclear translocation. On one hand, we used BHB addition at 400 μ M to mimic BHB overexpression in primary hepatocytes. As expected, BHB treatment blocked glucagon-induced nuclear translocation of FOXO1 (Fig. 4c) and activated glucagon-inhibited FOXO1 phosphorylation (Fig. 4e). In addition, acetylated-lysine antibody of FOXO1

showed that BHB addition partly restored HADHA knockdown-attenuated acetylation upon glucagon challenge (Fig. 4g), indicating BHB inhibited FOXO1 deacetylation.

On the other hand, we induced BHB deficiency by knockdown of BDH1. Of note, BDH1 catalyzes the metabolic transformation of AcAc to BHB as the last enzyme of hepatic ketogenesis, and can also reversibly transform BHB to AcAc as the first enzyme of ketolysis (Annual Review of Nutrition, 2017, 37: 51-76). Ketogenesis is greatest in the liver, the main ketogenic organ, about 10-fold higher than other organs (Journal Inherited Metabolic Disease, 2020, 43(5): 960-968). In contrast, ketolysis that generally takes place in extra-hepatic tissues cannot occur in the liver because of lack of ketone body oxidation enzyme OXCT1/SCOT (Annual Review of Nutrition, 2017, 37: 51-76). Hepatic BDH1 knockdown predominantly diminished circulating BHB levels, and at the same time considerably increased AcAc levels (Molecular Metabolism, 2021, 53:101269). In line with this, our results showed that BDH1 knockdown significantly inhibited HADHA-induced BHB release in primary hepatocytes challenged with glucagon (Supplementary Fig. 10b), and diminished the effects of HADHA overexpression on BHB production in the serum (Fig. 7b) and the liver (Fig. 7c) of HFD-fed mice. BDH1 knockdown diminished the inhibitory effect of HADHA on nuclear translocation of FOXO1 (Fig. 4b), and antagonized HADHA-activated FOXO1 phosphorylation in primary hepatocytes challenged with glucagon (Fig. 4d). In addition, acetylated-lysine antibody of FOXO1 showed that BHB knockdown antagonized HADHA-enhanced acetylation in response to glucagon challenge (Fig. 4f), indicating BDH1 deficiency enhanced FOXO1 deacetylation. In close agreement, in HFD-fed mice, BDH1 deficiency enhanced FOXO1 expression both in the nucleus and cytoplasm in response to HADHA overexpression (Fig. 7h). HADHA induction preserved FOXO1 acetylation and inactivated FOXO1 by increasing phosphorylation, but this effect was diminished by BDH1 knockdown (Fig. 7i, j).

We have added the relevant information (Lines 191-193, 216-228 and 286-300) and figures (Fig. 4, Supplementary Fig. 10b and Fig. 7) in the Revised Manuscript.

Fig. 4 HADHA and BHB inhibited glucagon-induced FOXO1 activation. **a** Luciferase reporter assay of the inhibitory effect of HADHA on *Pck1* and *G6pc* gene promoter co-transfected with HADHA and FOXO1 plasmid in 293T cells. The luciferase activity was normalized with the internal control (Renilla luciferase, $n = 6$). **b** Representative confocal images of FOXO1 nuclear translocation in HADHA-overexpressed primary hepatocytes with or without BDH1 siRNA transfection and 100 nM glucagon stimulation for 1 h. Scale bar represents 10 μm . **c** Confocal images of FOXO1 nuclear translocation in BHB (400 μM , 6 h) treated primary hepatocytes when exposed to glucagon (100 nM, 1 h). Scale bar represents 10 μm . **d** FOXO1 phosphorylation from the hepatocytes in panel **b** ($n = 3$). **e** FOXO1 phosphorylation from the hepatocytes in panel **c** ($n = 3$). **f**, **g** FOXO1 acetylation in HADHA-overexpressed or knockdown primary hepatocytes transfected with BDH1 siRNA and 100 nM glucagon stimulation for 1 h ($n = 3$). BHB β -hydroxybutyrate, GLC glucagon, NC normal control. Values represent mean \pm SEM. Statistical differences were determined by one-way ANOVA. $**p < 0.01$ vs. the control group. $###p < 0.01$ vs. the indicated treatment. Source data are provided as a Source Data file.

Supplementary Fig. 10b BHB level in HADHA-overexpressed primary hepatocytes transfected with BDH1 siRNA, 100 nM glucagon stimulation for 1 h (n = 6). BDH1 β -hydroxybutyrate dehydrogenase 1, BHB β -hydroxybutyrate, GLC glucagon. Bars represent mean \pm SEM values. Statistical differences were determined by one-way ANOVA. ** $p < 0.01$ vs. the control group. ## $p < 0.01$ vs. indicated treatments. Source data are provided as a Source Data file.

Fig. 7 (b, c) Eight-week-old mice injected with AAV8-NC, AAV8-HADHA or AAV8-BDH1 shRNA were fed with NCD or HFD for 12 weeks. BHB contents in serum and liver (n = 4-5). AAV adeno-associated virus, BDH1 β -hydroxybutyrate dehydrogenase 1, BHB β -hydroxybutyrate, HFD high-fat diet, NCD normal chow diet. Values represent mean \pm SEM. Statistical differences were determined by one-way ANOVA. ** $p < 0.01$ vs. the control group. ## $p < 0.01$ vs. the indicated treatments. Source data are provided as a Source Data file.

Fig. 7 (h-j) Eight-week-old mice injected with AAV8-NC, AAV8-HADHA or AAV-BDH1 shRNA were fed with NCD or HFD for 12 weeks. **h** Protein expressions of FOXO1, HDAC4, HDAC5 and HDAC7 in nucleus and cytoplasm of liver (n = 3). **i** Hepatic FOXO1 acetylation levels. **j** Western blotting analysis of HADHA and FOXO1 phosphorylation in the liver (n = 3). AAV adeno-associated virus, AUC area under the curve, BDH1 β-hydroxybutyrate dehydrogenase 1, BHB β-hydroxybutyrate, HFD high-fat diet, NCD normal chow diet. Values represent mean ± SEM. Statistical differences were determined by one-way ANOVA. * $p < 0.05$, ** $p < 0.01$ vs. the control group. # $p < 0.05$, ### $p < 0.01$ vs. the indicated treatments. Source data are provided as a Source Data file.

2) Page 8, line 171: It is surprising that BDH1 knockdown (Supplementary Fig. 4a) blocked the HADHA overexpression-inhibited hepatic glucagon response in gluconeogenesis-associated genes and hepatic glucose production (Fig. 3e, f). How does this occur? Using AcAc, BDH1 increased expression of these genes. It suggests that BHB itself is important for expression of the gluconeogenic enzymes. Therefore, preventing BHB conversion to AcAc by knocking down BDH1 does not make sense.

Response: We think our point raised in line 171, page 8 of our previous manuscript was misunderstood by the reviewer probably due to lack of clarity. Knockdown of BDH1 in the liver prevented the conversion of AcAc to BHB rather than preventing conversion of BHB to AcAc. Of note, BDH1 catalyzes the metabolic conversion of AcAc to BHB as the last enzyme of hepatic ketogenesis, and can also reversibly convert BHB to AcAc

as the first enzyme of ketolysis in extrahepatic tissues (Annual Review of Nutrition, 2017, 37: 51-76). Ketogenesis mainly takes place in the liver (Journal Inherited Metabolic Disease, 2020, 43(5): 960-968), while ketolysis can only occur in extrahepatic tissues because of the lack of ketone body oxidation enzyme OXCT1/SCOT in hepatocytes (Annual Review of Nutrition, 2017, 37: 51-76). Hepatic BDH1 knockdown predominantly diminished circulating BHB levels, and at the same time considerably increased AcAc levels (Molecular Metabolism, 2021, 53:101269). In line with this, our results showed that BDH1 knockdown significantly inhibited HADHA-induced BHB release in primary hepatocytes challenged with glucagon (Supplementary Fig. 10b), and diminished HADHA-induced BHB levels in the serum (Fig. 7b) and the liver (Fig. 7c) of HFD-fed mice.

We have shown that BDH1 knockdown inhibited BHB production, and BHB is a negative regulator of hepatic glucose production. It is thus not surprising to observe that BDH1 knockdown blocked the HADHA-inhibited hepatic glucagon response in gluconeogenesis-associated genes and hepatic glucose production (Fig. 3e, f). These findings can be attributable to BHB deficiency.

To compare the separate roles of two ketone bodies regulating gluconeogenesis, we knocked down BDH1 in hepatocytes to prevent interconversion of the two ketones, and then add BHB or AcAc separately. Interestingly, BHB could still effectively inhibit glucagon-induced gluconeogenesis, but AcAc failed to produce inhibitory effects in the BDH1-knockdown hepatocytes (Supplementary Fig. 11). Together, BHB, but not AcAc, is responsible for the inhibitory effects of HADHA on hepatic gluconeogenesis.

We have added the relevant information (Lines 191-193, 206-208 and 286-290) and figures (Fig. 3e-f, Supplementary Fig. 10b, Supplementary Fig. 11 and Fig. 7b-c) in the Revised Manuscript.

Supplementary Fig. 10b BHB level in HADHA-overexpressed primary hepatocytes transfected with BDH1 siRNA, 100 nM glucagon stimulation for 1 h (n = 6). BDH1 β -hydroxybutyrate dehydrogenase 1, BHB β -hydroxybutyrate, GLC glucagon. Bars represent mean \pm SEM values. Statistical differences were determined by one-way ANOVA. ** $p < 0.01$ vs. the control group. ### $p < 0.01$ vs. indicated treatments. Source data are provided as a Source Data file.

Fig. 7 (b, c) Eight-week-old mice injected with AAV8-NC, AAV8-HADHA or AAV8-BDH1 shRNA were fed with NCD or HFD for 12 weeks. BHB contents in serum and liver (n = 4-5). AAV adeno-associated virus, BDH1 β -hydroxybutyrate dehydrogenase 1, BHB β -hydroxybutyrate, HFD high-fat diet, NCD normal chow diet. Values represent mean \pm SEM. Statistical differences were determined by one-way ANOVA. ** $p < 0.01$ vs. the control group. ### $p < 0.01$ vs. the indicated treatments. Source data are provided as a Source Data file.

Fig. 3 (e, f) e Relative mRNA abundance of *Pck1*, *G6pc* and *Pgcl1a* in HADHA-overexpressed primary hepatocytes transfected BDH1 siRNA with or without BHB pretreatment (400 μ M, 6 h), 100 nM glucagon stimulation for 1 h (n = 6). f HGP from the hepatocytes in panel e (n = 6). BDH1 β -hydroxybutyrate dehydrogenase 1, BHB β -hydroxybutyrate, GLC glucagon, HGP hepatic glucose production. Values represent mean \pm SEM. Statistical differences were determined by one-way ANOVA. ** $p < 0.01$

vs. the control group. $^{###}p < 0.01$ vs. indicated treatments. Source data are provided as a Source Data file.

Supplementary Fig. 11 BHB, but not AcAc inhibited gluconeogenesis. Relative mRNA abundance of *Pck1*, *G6pc* and *Pgc1a* in primary hepatocytes transfected with BDH1 siRNA in the presence of BHB or AcAc (400 μ M, 6 h), 100 nM glucagon stimulation for 1 h (n = 6). AcAc acetoacetate, BDH1 β -hydroxybutyrate dehydrogenase 1, BHB β -hydroxybutyrate, GLC glucagon, ns no significance, PBS phosphate buffer solution. Bars represent mean \pm SEM values. Statistical differences were determined by one-way ANOVA. $^{**}p < 0.01$ vs. the control group. Source data are provided as a Source Data file.

3) BDH1 is a mitochondrial enzyme important in BHB oxidation. However, knocking down BDH1 decreased the expression of gluconeogenic enzymes. One would expect that knocking down BDH1 would increase BHB available for nuclear HDAC7 interaction, and therefore decrease gluconeogenic enzyme transcription. This possibility is not addressed.

Response: This comment is similar to the previous one. As we explained already, we feel the reviewer misunderstood the information we sought to convey. Actually, our results (Supplementary Fig. 10b, Fig. 7b and 7c), together with others (Journal Inherited Metabolic Disease, 2020, 43(5): 960-968; Molecular Metabolism, 2021, 53:101269), clearly showed that knocking down BDH1 limited BHB production from its upstream metabolite AcAc rather than increasing BHB availability for nuclear HDAC7 interaction. One main reason is that hepatic BDH1 mainly converts AcAc to BHB as the last step during ketogenesis, because ketolysis cannot occur in the liver due to the

lack of ketone body oxidation enzyme OXCT1/SCOT. As a result, knockdown of BDH1 decreased the expression of gluconeogenic enzymes because of a sharp decline in BHB availability for nuclear HDAC7 inactivation (Fig. 5a and Supplementary Fig. 12a). We have added the relevant information (Lines 191-193, 236-239 and 286-290) and figures (Supplementary Fig. 10b, Fig. 5a, Supplementary Fig. 12a and Fig. 7b-c) in the Revised Manuscript.

Supplementary Fig. 10b BHB level in HADHA-overexpressed primary hepatocytes transfected with BDH1 siRNA, 100 nM glucagon stimulation for 1 h (n = 6). BDH1 β-hydroxybutyrate dehydrogenase 1, BHB β-hydroxybutyrate, GLC glucagon. Bars represent mean ± SEM values. Statistical differences were determined by one-way ANOVA. ***p* < 0.01 vs. the control group. ##*p* < 0.01 vs. indicated treatments. Source data are provided as a Source Data file.

Fig. 7 (b, c) Eight-week-old mice injected with AAV8-NC, AAV8-HADHA or AAV8-BDH1 shRNA were fed with NCD or HFD for 12 weeks. BHB contents in serum and liver (n = 4-5). AAV adeno-associated virus, BDH1 β-hydroxybutyrate dehydrogenase 1, BHB β-hydroxybutyrate, HFD high-fat diet, NCD normal chow diet. Values represent mean ± SEM. Statistical differences were determined by one-way ANOVA. ***p* < 0.01 vs. the control group. ##*p* < 0.01 vs. the indicated treatments. Source data are provided as a Source Data file.

Fig. 5a The distribution of HDAC4, HDAC5 and HDAC7 in nucleus and cytoplasm of HADHA-overexpressed hepatocytes transfected with BDH1 siRNA with or without BHB (400 μ M, 6 h) and 100 nM glucagon stimulation for 1 h (n = 3). BDH1 β -hydroxybutyrate dehydrogenase 1, BHB β -hydroxybutyrate, GLC glucagon, NC normal control. Values represent mean \pm SEM. Statistical differences were determined by one-way ANOVA. ** p < 0.01 vs. the control group. ### p < 0.01 vs. indicated treatments. Source data are provided as a Source Data file.

Supplementary Fig. 12a Representative confocal image of primary hepatocytes transfected with HADHA plasmid with or without BDH1 siRNA, 100 nM glucagon stimulation for 1 h. BDH1 β -hydroxybutyrate dehydrogenase 1, BHB β -hydroxybutyrate, GLC glucagon, NC normal control.

4) It is not clear how downregulation of HADHA decrease ketogenesis. Normally, increases in fatty acid oxidation result in increased ketogenesis. This is opposite of what is expected, since HADHA is a key enzyme complex of fatty acid oxidation.

Response: Our results actually are in agreement with the reviewer's comment in respect of the fact that increase in fatty acid oxidation results in increased ketogenesis. Fatty acid β -oxidation-derived acetyl-CoA is the main source of ketogenesis. Mitochondrial

trifunctional protein (MTP) catalyzes three reactions in the fatty acid β -oxidation, and HADHA is a main subunit of MTP required for enzymatic activation. During starvation, fat mobilization increases hepatic mitochondrial β -oxidation to provide acetyl-CoA for ketone body production. Thus, HADHA overexpression enhances fatty acid oxidation and correspondingly promotes downstream ketogenesis (Fig. 2b-d). On the contrary, downregulation of HADHA decreases fatty acid oxidation and thereby reduces ketone body production (Fig. 2e-g). We reorganized the manuscript to make it easier for readers to follow. We have added the relevant information (Lines 157-167) and figure (Fig. 2) in the Revised Manuscript.

Fig. 2 Stable isotope tracing of ketogenesis. **a** A schematic summary of isotope tracing of ketogenesis using $[U-^{13}C]$ palmitate. **(b-d)** Primary hepatocytes treated with $[U-^{13}C]$ palmitate (0.1 mM) for 4 h following 100 nM glucagon stimulation for 1 h with or without HADHA plasmid transfection ($n = 3-8$). Acetyl-CoA serves as a precursor of ketogenesis or TCA cycle. **b** BHB relative abundance. **c** AcAc relative abundance. **d** Cit relative abundance. **(e-g)** Primary hepatocytes treated with $[U-^{13}C]$ palmitate (0.1 mM) for 4 h following 100 nM glucagon stimulation for 1 h with or without HADHA siRNA transfection ($n = 3-5$). **e** BHB relative abundance. **f** AcAc relative abundance. **g** Cit relative abundance. AcAc acetoacetate, BHB β -hydroxybutyrate, Cit citrate, GLC glucagon, TCA tricarboxylic acid cycle. Bars represent mean \pm SEM values. Statistical differences were determined by one-way ANOVA. * $p < 0.05$, ** $p < 0.01$ vs. the control group; # $p < 0.05$, ## $p < 0.01$ vs. indicated treatments. Source data are provided as a Source Data file.

5) It is proposed that glucagon inhibition of HADHA decreases acetyl CoA production from fatty acid oxidation. Evidence to support this is weak. Furthermore, what is happening to this acetyl CoA is not clear. Is it being used for FoxO1 acetylation? Data

to support this is not provided. Furthermore, it is not clear how the acetyl CoA would be getting out of the mitochondria to alter nuclear acetylation.

Response: We added an experiment to address the reviewer's comment on acetyl-CoA production. Glucagon impaired HADHA, a key enzyme in regulating fatty acid β -oxidation, and inhibited ketogenesis due to reduced acetyl-CoA fuel from β -oxidation. As expected, results showed glucagon stimulation significantly decreased acetyl-CoA levels in hepatocytes, while HADHA overexpression increased acetyl-CoA under physiological conditions and reversed glucagon-inhibited acetyl-CoA level (Supplementary Fig. 5). We have added the relevant information (Lines 150-153) and figure (Supplementary Fig. 5) in the Revised Manuscript.

In this work, HADHA-mediated acetyl-CoA production from β -oxidation serves as the substrate for ketone body production. Whether or not mitochondrial acetyl-CoA as acetyl donor is involved in FOXO1 acetylation remains to be explored. We speculated that it is possible that mitochondrial acetyl-CoA donates acetyl for FOXO1 acetylation in the nucleus. Mitochondrial acetyl-CoA serves as an important source for providing acetyl donors for protein acetylation such as FOXO1 acetylation in cytosol and nucleus. Acetyl-CoA ATP-citrate lyase (ACLY) is responsible for interconversion of acetyl-CoA to citrate by shuttling between the cytosol and nucleus. ACLY plays a key role in assisting acetyl-CoA to get out of the mitochondria into the cytoplasm and then translocate into the nucleus (Science, 2009, 324: 1076-1080). Since this aspect is out of the aims of this study, we did not mention this in the text.

Supplementary Fig. 5 HADHA promoted acetyl-CoA production. Acetyl-CoA levels in HADHA-overexpressed primary hepatocytes with or without 100 nM glucagon stimulation for 1 h (n = 5). GLC glucagon. Bars represent mean \pm SEM values. Statistical differences were determined by one-way ANOVA. ** $p < 0.01$ vs. the control group, ## $p < 0.01$ vs. indicated treatments. Source data are provided as a Source Data

file.

6) Figure 1g: It appears that overexpression of HADHA in liver increases G6Pase and PGC-1a, and marginally increased PEPCK in mice compared to controls. Why? This suggests that there is some increase in gluconeogenesis in HADHA overexpressing mice. This is not what would be expected if HDADA deficiency is increasing gluconeogenesis.

Response: We believe the reviewer might misunderstand the effects of liver-specific HADHA overexpression in mice. Our results showed that HADHA overexpression on gluconeogenesis significantly inhibited gluconeogenesis-associated genes expression such as *Pck1*, *G6pc* and *Pgc1a* in response to glucagon challenge (Fig. 1g). In contrast, HADHA knockdown directly enhanced gluconeogenic genes expression and aggravated glucagon-induced genes expression in the liver of mice (Fig. 1d). These results were replicated in primary hepatocytes (Supplementary Fig. 3).

Fig. 1g mRNA levels of *Pck1*, *G6pc* and *Pgc1a* in the livers of mice that were either treated with AAV8-HADHA or AAV8-NC after glucagon injection (2 mg/kg body weight, n = 5). AAV adeno-associated virus, GLC glucagon, NC normal control. Bars represent mean \pm SEM values. Statistical difference between two groups were determined by a two-tailed Student's *t* test, and all others were used one-way ANOVA. ** $p < 0.01$ vs. the control group. ### $p < 0.01$ vs. indicated treatments. Source data are provided as a Source Data file.

Fig. 1d mRNA levels of *Pck1*, *G6pc* and *Pgc1a* in the livers of the mice subjected to glucagon challenge (2 mg/kg) and treated with AAV8-HADHA shRNA or AAV8-NC (n = 5). AAV adeno-associated virus, GLC glucagon, NC normal control. Bars represent mean \pm SEM values. Statistical difference between two groups were determined by a two-tailed Student's *t* test, and all others were used one-way ANOVA. **p* < 0.05, ***p* < 0.01 vs. the control group. #*p* < 0.05, ##*p* < 0.01 vs. indicated treatments. Source data are provided as a Source Data file.

Supplementary Fig. 3 HADHA inhibited hepatic glucagon response in vitro. **a** mRNA levels of *Hadha* siRNA sequences in primary hepatocytes (n = 6). **b** Relative mRNA abundance of *Pck1*, *G6pc* and *Pgc1a* in primary hepatocytes stimulated by glucagon (100 nM, 1 h) after HADHA knockdown (n = 6). **c** HGP from primary hepatocytes in panel **b** (n = 6). **d** mRNA levels of *Hadha* in HADHA-overexpressed primary hepatocytes. **e** Relative mRNA abundance of *Pck1*, *G6pc* and *Pgc1a* in primary hepatocytes stimulated by glucagon (100 nM, 1 h) after HADHA overexpression (n = 6). **f** HGP from primary hepatocytes in panel **e** (n = 6). AAV adeno-associated virus, GLC glucagon, HGP hepatic glucose production, NC normal control. Bars represent mean \pm SEM values. Statistical differences were determined by one-way ANOVA. **p* < 0.05, ***p* < 0.01 vs. the control group. Source data are provided as a Source Data file.

7) Page 7, line 147: How was ¹³C-palmitic acid added to hepatocytes? A concentration of 0.1 mM was used. Was it bound to albumin and what was the albumin concentration used?

Response: We apologize for the lack of adequate experimental details. We used bovine serum albumin (BSA) at a concentration of 40% with the reagent buffer to conjugate

[U-¹³C]palmitate (20 mM). A final concentration of 0.1 mM conjugated [U-¹³C]palmitate was added to the primary hepatocytes and incubated for 4 h. This information has been detailed in the Methods (Lines 453-457) and Figure Legends (Lines 725-728) in the Revised Manuscript.

8) Page 7: What happens to label incorporation in citrate from ¹³C-palmitate when HADHA is decreased?

Response: As requested by the reviewer, ketone bodies in HADHA-knockdown hepatocytes were also analyzed by stable isotope tracing and mass spectrometry. [U-¹³C]palmitate (0.1 mM) was added to primary hepatocytes and incubated for 4 h. BHB (M2) and BHB (M4) decreased by 40% and 30% respectively after glucagon stimulation (Fig. 2e). AcAc (M2) and AcAc (M4) decreased by 35% and 40% (Fig. 2f). It is worth noting that HADHA knockdown further inhibited ketogenesis, BHB (M2) and BHB (M4) by 45%, AcAc (M2) by 50%, and AcAc (M4) by 40% (Fig. 2e, f). HADHA knockdown also inhibited the production of Cit (M2) derived from [U-¹³C]palmitate (Fig. 2g), and augmented glucagon-inhibited Cit (Fig. 2f). Related results (Lines 165-167) and data (Fig. 2e-g) have been added in the Revised Manuscript.

Fig. 2 (e-g) Primary hepatocytes treated with [U-¹³C]palmitate (0.1 mM) for 4 h following 100 nM glucagon stimulation for 1 h with or without HADHA siRNA transfection (n = 3-5). **e** BHB relative abundance. **f** AcAc relative abundance. **g** Cit relative abundance. AcAc acetoacetate, BHB β-hydroxybutyrate, Cit citrate, GLC glucagon. Bars represent mean ± SEM values. Statistical differences were determined by one-way ANOVA. **p* < 0.05, ***p* < 0.01 vs. the control group; #*p* < 0.05, ###*p* < 0.01 vs. indicated treatments. Source data are provided as a Source Data file.

9) Figure 2D: Glucagon did not increase ACAC [M+2] abundance. This is not consistent with glucagon decreasing fatty acid acetyl CoA going to ketones.

Response: As the reviewer indicated, glucagon stimulation showed a no significant but

a trend in inhibiting AcAc (M2) abundance by isotope tracing (Fig. 2d of the previous version of our manuscript). Of note, glucagon significantly decreased AcAc (M4) level (Fig. 2d of the previous version of our manuscript). Based on the suggestions of Reviewer 1# and Reviewer 4#, we combined the former Fig. 2 and Supplementary Fig. 3 into a novel figure (Fig. 2) in the Revised Manuscript. Four groups were presented in the same graphs, including control group, HADHA overexpression group, glucagon stimulation group, and HADHA overexpression plus glucagon group (Fig. 2 in the Revised Manuscript).

Furthermore, both labeled metabolites and unlabeled fractions were determined by LC-MS/MS analysis. Unlabeled metabolite fraction for AcAc (M0) further confirmed a significant inhibition effect of glucagon on AcAc production (Fig. 2c in the Revised Manuscript).

In addition, we also studied the role of glucagon in AcAc production in HADHA-knockdown hepatocytes using stable isotope tracing combined with LC-MS/MS analysis. In close agreement, the level of unlabeled AcAc fraction (M0), and labeled AcAc (M2 and M4) were simultaneously decreased in response to glucagon stimulation (Fig. 2f in the Revised Manuscript).

These findings, together with the results of another ketone body BHB (Fig. 2), support our hypothesis that glucagon decreased ketone body derived from fatty acid oxidation by impairing HADHA.

Fig. 2 Stable isotope tracing of ketogenesis. a A schematic summary of isotope tracing of ketogenesis using [U-¹³C]palmitate. (b-d) Primary hepatocytes treated with

[U-¹³C]palmitate (0.1 mM) for 4 h following 100 nM glucagon stimulation for 1 h with or without HADHA plasmid transfection (n = 3-8). Acetyl-CoA serves as a precursor of ketogenesis or TCA cycle. **b** BHB relative abundance. **c** AcAc relative abundance. **d** Cit relative abundance. **(e-g)** Primary hepatocytes treated with [U-¹³C]palmitate (0.1 mM) for 4 h following 100 nM glucagon stimulation for 1 h with or without HADHA siRNA transfection (n = 3-5). **e** BHB relative abundance. **f** AcAc relative abundance. **g** Cit relative abundance. AcAc acetoacetate, BHB β-hydroxybutyrate, Cit citrate, GLC glucagon, TCA tricarboxylic acid cycle. Bars represent mean ± SEM values. Statistical differences were determined by one-way ANOVA. **p* < 0.05, ***p* < 0.01 vs. the control group; #*p* < 0.05, ##*p* < 0.01 vs. indicated treatments. Source data are provided as a Source Data file.

10) Figure 3g: Data on the effects of BDH1 knockdown on AcAc effects on hepatic glucose production are missing. Why wasn't this presented?

Response: Sorry for the missing data. We have performed new experiments to investigate the effects of AcAc on hepatic glucose production with or without BDH1 knockdown. The results showed that AcAc inhibited hepatic glucose production (Fig. 3h). However, BDH1 knockdown diminished the effects of AcAc in hepatic glucose production (Fig. 3h). Related results (Line 200) and data (Fig. 3h) have been added to the Revised Manuscript.

Fig. 3h HGP from the AcAc pretreated (400 μM) primary hepatocytes with or without BDH1 siRNA transfection (n = 6). AcAc acetoacetate, BDH1 β-hydroxybutyrate dehydrogenase 1, GLC glucagon, HGP hepatic glucose production, PBS phosphate buffer solution. Values represent mean ± SEM. Statistical differences were determined by one-way ANOVA. ***p* < 0.01 vs. the control group. ##*p* < 0.01 vs. indicated treatments. Source data are provided as a Source Data file.

11) Figure 4: Where is HADHA expressed or overexpressed? Is it in mitochondria? How does mitochondrial HADHA inhibit nuclear translocation of FoxO1 (or nuclear transcription)?

Response: HADHA is mainly located in the mitochondria (data from Cell atlas and

Genecards). Our work showed that HADHA exerts its function in the mitochondria to increase BHB production by promoting mitochondrial β -oxidation. Ketone bodies such as BHB are small molecules that freely diffuse through the mitochondrial membrane and nuclear pore complex (Science, 2009, 324: 1076-1080). BHB acted as a signaling metabolite to selectively inhibit HDAC7 activity via interaction with the residue of Glu543. BHB, by HDAC7 inactivation, preserved FOXO1 acetylation to facilitate FOXO1 nuclear exclusion, resultantly inhibiting hepatic gluconeogenesis.

The related information about the role of HADHA (Fig. 8) has been added in the Revised Manuscript.

(<https://v18.proteinatlas.org/ENSG00000084754-HADHA/cell>)

(<https://www.genecards.org/cgi-bin/carddisp.pl?gene=HADHA&keywords=HADHA>)

12) Figure 5: HADHA overexpression decreases glucagon effects on nuclear HDAC7.

How does this occur? There is no data on what HADHA overexpression does to nuclear HDAC7 in the absence of glucagon.

Response: This work showed that HADHA inhibited hepatic glucagon response not by directly interacting with HDAC7, but through ketone body BHB as the mediator. Specifically, HADHA overexpression increased BHB generation by promoting mitochondrial β -oxidation. As a small molecule, BHB is able to freely diffuse through the mitochondrial membrane and nuclear pore complex (Science, 2009, 324: 1076-1080). Elevated BHB selectively inhibited HDAC7 activity. BHB inhibition of HDAC7 preserved FOXO1 acetylation to facilitate FOXO1 nuclear exclusion.

As requested by the reviewer, we have performed new experiments regarding the effects of HADHA overexpression on nuclear HDAC7 in the absence of glucagon (Fig. X3). The results showed that HADHA overexpression decreased nuclear translocation of HDAC7, but not translocation of HDAC4 and 5 in the absence or presence of glucagon stimulation.

Fig. X3. Immunoprecipitation analysis the distribution of HDAC4, 5 and 7 in the nucleus and cytoplasm. The distribution of HDAC4, 5 and 7 in nucleus and cytoplasm of HADHA-overexpressed hepatocytes transfected with BDH1 siRNA with or without BHB (400 μ M), 1 h after 100 nM glucagon stimulation (n = 3). GLC glucagon, ns no significance, PBS phosphate buffer solution. Values represent mean \pm SEM. Statistical differences were determined by one-way ANOVA. * p < 0.05, ** p < 0.01 vs. the control group. # p < 0.05, ## p < 0.01 vs. indicated treatments.

13) No data is provided as to what these procedures are actually doing to acetyl CoA levels or ketone levels.

Response: As requested by the reviewer, we have detected acetyl-CoA level in

HADHA-overexpressed primary hepatocytes. Results showed glucagon stimulation significantly decreased acetyl-CoA levels in hepatocytes, which was rescued by HADHA overexpression (Supplementary Fig. 5).

Meanwhile, glucagon sharply decreased BHB level in hepatocytes. BHB levels inhibited by glucagon was restored by HADHA overexpression, and this effect was abolished when co-transfected with BDH1 (Supplementary Fig. 10b). These results indicated that HADHA promoted BHB production depending on BDH1. In addition, we have detected the serum BHB level in liver-specific HADHA overexpressed or knockdown mice. The results are in accordance with the in vitro data (Supplementary Fig. 6b, c).

We have added the relevant information (Lines 150-153, 170-173 and 191-193) and figures (Supplementary Fig. 5, Supplementary Fig. 6b-c and Supplementary Fig. 10b) in the Revised Manuscript.

Supplementary Fig. 5 HADHA promoted acetyl-CoA production. Acetyl-CoA levels in HADHA-overexpressed primary hepatocytes with or without 100 nM glucagon stimulation for 1 h (n = 5). GLC glucagon. Bars represent mean \pm SEM values. Statistical differences were determined by one-way ANOVA. ** $p < 0.01$ vs. the control group, ### $p < 0.01$ vs. indicated treatments. Source data are provided as a Source Data file.

Supplementary Fig. 10b BHB level in HADHA-overexpressed primary hepatocytes transfected with BDH1 siRNA and stimulated with 100 nM glucagon stimulation for 1 h (n = 6). BDH1 β -hydroxybutyrate dehydrogenase 1, BHB β -hydroxybutyrate, GLC

glucagon. Bars represent mean \pm SEM values. Statistical differences were determined by one-way ANOVA. $**p < 0.01$ vs. the control group. $##p < 0.01$ vs. indicated treatments. Source data are provided as a Source Data file.

Supplementary Fig. 6 (b, c) Serum BHB levels in HADHA liver-specific overexpression (b) or knockdown (c) mice with or without glucagon stimulation (2 mg/kg, n = 5). AAV adeno-associated virus, BHB β -hydroxybutyrate, GLC glucagon, NC normal control. Values represent mean \pm SEM. Statistical differences were determined by one-way ANOVA. $*p < 0.05$, $**p < 0.01$ vs. the control group. $##p < 0.01$ vs. indicated treatments. Source data are provided as a Source Data file.

14) Figure 6 b and c: Why are serum levels of BHB presented as relative levels? Why not absolute levels?

Response: We apologize for the mistake. We provided absolute levels for BHB as $\mu\text{mol/L}$ in serum and as $\mu\text{mol/mg prot}$ in the liver normalized by the tissue protein content (Fig. 7b, c).

Fig. 7 (b, c) Eight-week-old mice injected with AAV8-NC, AAV8-HADHA or AAV8-BDH1 shRNA were fed with NCD or HFD for 12 weeks. BHB contents in serum and liver (n = 4-5). AAV adeno-associated virus, BDH1 β -hydroxybutyrate dehydrogenase 1, BHB β -hydroxybutyrate, HFD high-fat diet, NCD normal chow diet. Values represent mean \pm SEM. Statistical differences were determined by one-way ANOVA. $**p < 0.01$ vs. the control group. $##p < 0.01$ vs. the indicated treatments. Source data are provided as a Source Data file.

15) Figure 6d: When liver BDH1 was silenced, glucose tolerance was worsened.

Response: This comment is similar to previous queries by the reviewer. As earlier indicated, we believe the reviewer misunderstood the information we provided in respect of the effect of BDH1 knockdown on glucose metabolism. Actually, our results (Fig. 7b-c), together with previous reports (Journal Inherited Metabolic Disease, 2020, 43(5): 960-968; Molecular Metabolism, 2021, 53:101269), indicate that knockdown of BDH1 limited BHB production rather than increased BHB level in the serum (Fig. 7b) and in the liver of mice (Fig. 7c). Since BDH1 mediated the inhibitory effects of HADHA on hepatic glucagon response, BDH1 knockdown abolished the improvement of HADHA overexpression on pyruvate tolerance in HFD-fed mice (Fig. 7d).

Fig. 7 (b-d) Eight-week-old mice injected with AAV8-NC, AAV8-HADHA or AAV-BDH1 shRNA were fed with NCD or HFD for 12 weeks. **b, c** BHB contents in serum and liver (n = 4-5). **d** Pyruvate tolerance test (2 g/kg body weight) after 11 weeks of feeding. AUC is indicated on the right (n = 6). AAV adeno-associated virus, BDH1 β -hydroxybutyrate dehydrogenase 1, BHB β -hydroxybutyrate, HFD high-fat diet, NCD normal chow diet. Values represent mean \pm SEM. Statistical differences were determined by one-way ANOVA. ** $p < 0.01$ vs. the control group. ### $p < 0.01$ vs. the indicated treatments. Source data are provided as a Source Data file.

Minor:

Page 6, line 130: fasting blood “glucose”...

Response: We apologize for the mistake, which has now been corrected in the Revised Manuscript.

Reviewer #4 (Remarks to the Author):

In this manuscript Pan, et al. investigate novel mechanisms underlying hepatic gluconeogenesis (GNG). The primary focus of this work is on the dysregulation of HADHA in response to glucagon or hyperglucagonemia observed after HF-feeding. The authors further implicate the HADHA metabolite, BHB, as critical inhibitor of HDAC7. Subsequently, acetylation/cytosolic retention of FoxO1 is lost and GNG is unrestricted. The data set is intriguing and expansive in scope. The authors should be commended on the comprehensive mechanistic studies. However, the article is poorly written with little acknowledgment of prior studies, nor explanation as to why their results are in complete opposition (especially in regards to glucagon and BHB dynamics). While the experiments are expansive, details as to how they were conducted are lacking. Likewise, controls for many studies are missing, making interpretation of the results impossible. Together these issues (detailed below) dramatically diminish my enthusiasm for this work.

Response: We thank the reviewer for the positive comments on our manuscript.

- (1) We thoroughly revised the manuscript to make it easier for readers to follow.
- (2) Detailed experimental conditions have been provided in the Methods and Figure Legends for other researchers to replicate our results.
- (3) In line with your comments, we examined the dynamic changes of fat mobilization, fatty acid oxidation, circulating glucagon and BHB levels during fasting. Moreover, we added new experimental results to support our findings, rewrote the paper to address what is different from previous work, and provided plausible explanations for our findings.
- (4) We have added the control group in related experiments to strengthen our results. For Fig. 1, we included AAV-HADHA shRNA control for 1c and 1d. For Fig. 1f and g, AAV-HADHA alone control was added. For Fig. 2, we added the control of HADHA plasmid alone. For Fig 3, BHB alone control was added.
- (5) We have improved our experimental designs and results in the Revised Manuscript.

Major:

1. The primary weakness of this report is a superficial nod to past studies and a lack of discussion how such vastly different effects are possible.

Response: The references cited were increased from 23 to 41, with the aim of providing more information about the study background and comparing our findings with published studies.

a. Pharmacological activation of the glucagon receptor elevates BHB, yet no mention of this confounding data is made.

Response: We thank the reviewer for bringing this to our attention. We performed a series of experiments to support our hypothesis that glucagon significantly inhibited ketone body production (Fig. 2, Supplementary Fig. 6b-c and Supplementary Fig. 10b). These findings directly challenged the classical dogma that glucagon has been described to promote ketogenesis in the fasted state. We added discussions to explain possible causing factors in Introduction and Discussion in the Revised Manuscript.

In the Introduction, we added (Lines 90-100): “In addition to driving glucose production, glucagon has been proposed to increase fatty acid oxidation and the production of ketone bodies¹⁸. Glucagon increases BHB production when perfused with oleic acid in rat liver¹⁹. In type 1 diabetic patients, suppression of glucagon secretion by somatostatin prevented ketoacidosis after acute insulin withdrawal²⁰. However, this view is challenged by a recent study, which showed that blocking glucagon receptor lowered fasting and fed glycemic levels, but did not alter BHB levels, suggesting that glucagon receptor signaling is not essential for ketogenesis²¹. In fact, the relationships among glucagon action, fatty acid oxidation and ketogenesis are very complex and involve diverse metabolic pathways, thus, making it difficult to precisely distinguish the sequence of altered metabolism from observed phenotypes in vivo.”

In the Discussion, we added (Lines 314-334): “Both impaired insulin action and aberrantly elevated glucagon are responsible for diabetic hyperglycemia²⁹. Emerging studies have put a spotlight on the regulation of glucagon by β cells in the development of diabetes³⁰. To mimic hyperglucagonemia in diabetes, we treated mice with high dose

of glucagon (2 mg/kg)²¹, and found impaired HADHA expression and ketone body production in the liver. These data are different from the traditional view that glucagon stimulates fatty acid oxidation and ketogenesis¹⁸⁻²⁰. Of note, the fact that glucagon stimulates ketone body production was only observed when insulin signaling is blocked^{21,31}, but a direct relationship between glucagon and ketogenesis has not been well established. A possible explanation could be that besides elevated glucagon, insulin levels are low during prolonged fasting, facilitating adipose lipolysis and fatty acid availability for oxidation and ketone body generation. In line with this explanation, insulin receptor antagonist elevated serum BHB levels in fed mice, while blocking glucagon signaling had no effect on plasma BHB levels in insulin-sensitive mice³¹. Different from glucagon actions as a physiological response during fasting, hyperglucagonemia is a pathological insult in diabetes²⁷. Similar to our observation in mice subjected to glucagon challenge, injection of glucagon at a supraphysiologic dose (1 mg/kg) is also shown to impair ketone body production in fasted mice²¹. Moreover, we showed the direct impact of glucagon in isolated hepatocytes. Glucagon challenge impaired HADHA protein to suppress β -oxidation, well explaining the reduction in ketogenesis under pathological conditions.”

b. Glucagon secretion is more tightly coupled to amino acid and fatty acid exposure (post prandial) than to hypoglycemia. This makes the current study difficult to reconcile with the physiology.

Response: We thank the reviewer for this interesting question. Our work did not focus on glucagon secretion but investigated the role of HADHA and BHB in glucagon response. Compared with wide investigations on insulin secretion, much less is known about glucagon secretion. It is generally speculated that fasting or starvation increase glucagon levels for maintaining euglycemia, but hypoglycemia does not always associate with glucagon levels (Diabetes, 2020, 69: 532-541). As the reviewer commented, amino acids and fatty acids seem to have tight relationships with glucagon levels, though the results are debatable. Fatty acid is shown to stimulate glucagon secretion from pancreatic islets (Diabetologia, 2008, 51: 1689-1693), partly attributable

to activation of membrane GPCR (Trends in Pharmacological Science, 2012, 33: 374-381). However, some clinical observation found that increased fatty acids suppressed glucagon secretion (Journal of Clinical Investigation, 1974, 53: 1284-1289). It is believed that insulin secretion is tightly coupled to β cells, and glucagon secretion is closely associated with α cells. Thus, the metabolites which contribute to or are more tightly coupled to glucagon secretion remains to be explored. Related discussion (Lines 365-373) has been added into the Revised Manuscript.

In this work, we selected high dose of glucagon as a stimulus (2 mg/kg for mice and 100 nM for primary hepatocytes). For one reason, accumulating evidence from human and animal studies shows that plasma glucagon concentrations are abnormally elevated in obesity and diabetes (Diabetes 2005, 54: 757-764; American Journal of Physiology Endocrinology and Metabolism, 2003, 284: E671-E678). Dysregulation of the hepatic glucagon response induces excessive hepatic glucose production, contributing to hyperglycemia in diabetes (Journal of Clinical Investigation, 2012, 122: 4-12). For another reason, single glucagon stimulation is a good model to study the effects of glucagon response and its underlying molecular mechanisms (Diabetes, 2020, 69: 882-892). Related information has been added into the Revised Manuscript (Lines 110-111 and 315-318).

2. Experimental conditions are brief or lacking (i.e. doses, time of treatment, prandial state of the mice at tissue collection).

Response: We have thoroughly checked the Methods in the manuscript and provided details on the experimental conditions (both in vitro and in vivo).

3. Critical controls are routinely omitted. Several examples are included below, but this is not fully inclusive of all missing control groups.

- a. Fig 1c,d,f,g – no vehicle treated AAV-HADHA group
- b. Fig 2b-e – same
- c. Fig 3a-d – no BHB (mono-treatment) control

Response: We thank the reviewer for drawing our attention to the missing data. We have

added the data on the controls in the Revised Manuscript.

For Fig. 1, we included the AAV-HADHA shRNA control for 1c and 1d. Results showed that liver-specific HADHA knockdown alone enhanced fasting blood glucose and gluconeogenesis-associated genes expression in mice (Fig. 1c, d). Upon glucagon stimulation, HADHA knockdown further aggravated blood glucose levels and gluconeogenic genes expression (Fig. 1c, d).

For Fig. 1f and g, AAV-HADHA control was added. Results showed that liver-specific HADHA overexpression alone slightly decreased fasting blood glucose and gluconeogenic genes expression in mice (Fig. 1f, g). Upon glucagon stimulation, HADHA overexpression considerably decreased glucagon-induced blood glucose levels and gluconeogenic genes expression (Fig. 1f, g).

For Fig. 2, we added the HADHA plasmid alone control, and combined initial Supplementary Fig. 3 data with Fig. 2. Both stable isotope tracing and unlabeled metabolite fractions showed that HADHA overexpression induced production of downstream metabolites such as BHB, AcAc and Cit in primary hepatocytes (Fig. 2b-d).

For Fig 3a-c, BHB alone control was added. Results showed that BHB had no significant effect on fasting blood glucose and *Pck1* in mice, but a slight decrease in *G6pc* and *Pgc1a* (Fig. 3a, b). Upon glucagon challenge, BHB greatly inhibited blood glucose and gluconeogenesis-associated genes expression.

We have added the relevant information (Lines 127-134, 157-165 and 179-181) and figures (Fig. 1b-g, Fig. 2b-d and Fig. 3a, b) in the Revised Manuscript.

Fig. 1 HADHA inhibited glucagon-stimulated hepatic gluconeogenesis. **a** Hepatic HADHA protein level in glucagon-challenged mice (2 mg/kg, n = 3). **b** Fasting blood glucose of liver-specific HADHA knockdown mice (n = 5). **c** Blood glucose levels in normal mice subjected to glucagon challenge (2 mg/kg) and treated with AAV8-HADHA shRNA or AAV8-NC. AUC is indicated on the right (n = 5). **d** mRNA levels of *Pck1*, *G6pc* and *Pgcl1* in the livers of the mice in panel **c** (n = 5). **e** Fasting blood glucose of liver-specific HADHA-overexpressed mice (n = 5). **f** Blood glucose curve and AUC for mice treated with AAV8-HADHA or AAV8-NC after glucagon injection (2 mg/kg, n = 5). **g** mRNA levels of *Pck1*, *G6pc* and *Pgcl1* in the livers of the mice in panel **f**. AAV adeno-associated virus, AUC area under the curve, GLC glucagon, NC normal control. Bars represent mean \pm SEM values. Statistical difference between two groups were determined by a two-tailed Student's *t* test, and all others were used one-way ANOVA. **p* < 0.05, ***p* < 0.01 vs. the control group. #*p* < 0.05, ###*p* < 0.01 vs. indicated treatments. Source data are provided as a Source Data file.

Fig. 2 (b-d) Primary hepatocytes treated with [U-¹³C]palmitate (0.1 mM) for 4 h following 100 nM glucagon stimulation for 1 h with or without HADHA plasmid transfection (n = 3-8). Acetyl-CoA serves as a precursor to fuel the ketogenesis or TCA cycle. **b** BHB relative abundance. **c** AcAc relative abundance. **d** Cit relative abundance. AcAc acetoacetate, BHB β -hydroxybutyrate, Cit citrate, GLC glucagon. Bars represent mean \pm SEM values. Statistical differences were determined by one-way ANOVA. **p* < 0.05, ***p* < 0.01 vs. the control group; #*p* < 0.05, ###*p* < 0.01 vs. indicated treatments. Source data are provided as a Source Data file.

Fig. 3 (a, b) **a** Blood glucose levels in normal mice subjected to glucagon challenge (2 mg/kg) and treated with BHB (100 mg/kg). AUC is indicated on the right (n = 5). **b** mRNA levels of *Pck1*, *G6pc* and *Pgcl1* in the livers of the mice in panel **a** (n = 5). AUC area under the curve, BHB β -hydroxybutyrate, GLC glucagon. Values represent mean \pm SEM. Statistical differences were determined by one-way ANOVA. **p* < 0.05, ***p* < 0.01 vs. the control group. ###*p* < 0.01 vs. indicated treatments. Source data are provided

as a Source Data file.

4. Why is FOXO1 overexpressed in Fig5? No control for this variable nor rationale is included.

Response: In the IP assay, the target protein is usually overexpressed in cells by plasmid transfection for normalizing its levels in each group and for facilitating its interaction with other proteins. As a result, in Fig. 5b, we overexpressed FOXO1 for IP. Results showed that HADHA overexpression obviously reduced the binding of FOXO1 to HDAC7 under glucagon stimulation, but had no significant impact on HDAC4 and 5 (Fig. 5b, Supplementary Fig. 12d). The inhibitory effect of HADHA overexpression on FOXO1 and HDAC7 interaction was abolished by BDH1 knockdown (Fig. 5b). Similarly, BHB addition inhibited the binding of FOXO1 to HDAC7 rather than HDAC4 and 5 (Supplementary Fig. 12d).

In addition, we have performed another IP experiment without the overexpression of FOXO1 and got a similar result (Fig. 4X).

Fig. 5b Immunoprecipitation analysis of interaction of FOXO1 with HDAC4, HDAC5 or HDAC7 in HADHA-overexpressed hepatocytes with or without BDH1 siRNA transfection (n = 3). BDH1 β-hydroxybutyrate dehydrogenase 1, BHB β-hydroxybutyrate, GLC glucagon NC normal control.

Supplementary Fig. 12d Immunoprecipitation analysis of the interaction of FOXO1 with HDAC4, HDAC5 or HDAC7 in glucagon-challenged (100 nM, 1 h) hepatocytes with or without BHB addition (400 μM, 6 h, n = 3). BHB β-hydroxybutyrate, GLC glucagon.

Fig. X4. Immunoprecipitation analysis of interaction between FOXO1 and HDAC4, HDAC5 or HDAC7. Assay was performed in HADHA-overexpressed hepatocytes with or without BDH1 siRNA transfection and 100 nM glucagon stimulation for 1 h (n = 3). BDH1 β-hydroxybutyrate dehydrogenase 1, BHB β-hydroxybutyrate, GLC glucagon NC normal control.

5. Why were HepG2 cells used for only 2 experiments (fig 5)?

Response: Primary mouse hepatocytes were used in most cellular experiments. Because the commercially available kits for HDACs enzyme activity are only of human origin, not from mouse, we have to employ human HepG2 cells for these two experiments.

6. Confirmation of overexpression and/or knockdown (KD) should be included for all targets, but especially for those of studies detailed in Fig 6, where genetic manipulation is conducted prior to 12w of HF-feeding. mRNA and Protein levels should be confirmed at the time of experiment. Current data included in the supplemental data is unclear as to timing and KD is partial in most targets, this needs to be addressed.

Response: Thank you for the invaluable suggestion. We have confirmed all the overexpression and/or knockdown mRNA levels in mice before the experiments. These results are added in the Revised Manuscript (Supplementary Fig. 15b, c). In addition, we have performed western blotting to detect the protein levels, and the results showed that the target proteins were efficiently overexpressed or knocked down in mice (Supplementary Fig. 2b and Supplementary Fig. 15d). Accordingly, we have detailed the exact time of the in vivo experiments in the Methods and Figure Legends.

Supplementary Fig. 15 (b, c) Eight-week-old mice injected with AAV-NC, AAV-HADHA or AAV-BDH1 shRNA were fed with NCD or HFD for 12 weeks. **b, c** mRNA levels of *Hadha* and *Bdh1* in the liver, heart, kidney and muscle of mice before HFD feeding (n = 3). AAV adeno-associated virus, NC normal control, ns no significance. Values represent mean \pm SEM. Statistical differences were determined by one-way ANOVA. ** $p < 0.01$ vs. the control group. Source data are provided as a Source Data file.

Supplementary Fig. 2 Western blotting confirmed HADHA protein expression. a Western blotting analysis of HADHA in the liver of mice injected with AAV8-HADHA or AAV8-NC. **b** Hepatic HADHA expression of mice injected with AAV8-HADHA shRNA or AAV8-NC. AAV adeno-associated virus, NC normal control. Source data are provided as a Source Data file.

Supplementary Fig. 15d Eight-week-old mice injected with AAV8-NC, AAV8-HADHA or AAV-BDH1 shRNA were fed with NCD or HFD for 12 weeks. **c** Western blotting analysis of BDH1 in the liver of mice injected with AAV-NC or AAV-BDH1 shRNA. AAV adeno-associated virus, BDH1 β -hydroxybutyrate dehydrogenase 1, NC normal control.

Minor:

1. The introduction (and really the article in general) is poorly constructed. As written, it reads like a loosely associated collection of statements.

Response: We have thoroughly reorganized and rewritten the manuscript to better bring clarity and it easier for readers to follow. Dr Raphael N. Alolga, a native English speaker helped to edit our manuscript.

2. Gene/protein nomenclature is muddled. The authors should review the current mouse standards and revise.

Response: We have thoroughly checked the manuscript and effected the necessary corrections in the Revised Manuscript.

Table X1. Gene and protein name.

Symbol	Species	Gene name	Protein name (abbreviation)
HADHA	Mice	Hadha	HADHA
HADHB	Mice	Hadhb	HADHB
PEPCK	Mice	Pck1	PEPCK
G6PC	Mice	G6pc	G6PC
PGC-1 α	Mice	Pgc1a	PGC-1 α
BDH1	Mice	Bdh1	BDH1
HDAC4	Mice	Hdac4	HDAC4
HDAC5	Mice	Hdac5	HDAC5
HDAC7	Mice	Hdac7	HDAC7

β -ACTIN	Mice	Actb	β -ACTIN
FOXO1	Mice	Foxo1	FOXO1

Reviewers' Comments:

Reviewer #1:

Remarks to the Author:

The authors provided several new data to address most of my concerns, but the key concern still remains.

1. The authors nicely provided insulin and somatostatin data in the setting of HADH overexpression. However, there must be some miscommunication. The most important thing this reviewer asked to assess is whether HADH "knockdown" mice (in Fig.1) are insulin-resistant (due to fat accumulation), which can explain all the serum glucose levels and hepatic gluconeogenesis gene expression, regardless of the authors' proposed mechanisms. Insulin serum levels do not probe hepatic insulin sensitivity. The authors should directly measure pAkt or other markers of hepatic insulin sensitivity in HADH knockdown mice.
2. To assess the physiological relevance of BHB administration, how much BHB blood levels are elevated after exogenous BHB administration (100mg/kg) in Fig.3?
3. It is nice to see acetyl-coA level changes as new data. Can the authors show acetyl-coA labeling from ¹³C-palmitate?
4. In the method, I would use 10mM instead of 40% as the BSA concentration used for tracing so that the readers can quickly notice palmitate:albumin ratio is 2:1.

Reviewer #2:

Remarks to the Author:

Most of the concerns that I raised have been properly addressed. The quality of SPR data has been improved. HDAC5's data look good. The data of HDAC4 and HDAC7 are not as good, but acceptable. However, there remain two minor points.

1. Please provide Rmax values for SPR fitting results (Fig. 6a). These values are helpful in evaluating the fitting results when compared with Chi2.
2. For docking of HDAC4, please provide docking scores and make a comparison with the docking result of HDAC7.

Reviewer #3:

Remarks to the Author:

none

Reviewer #4:

Remarks to the Author:

The authors have responded to the concerns I presented and have improved the manuscript.

Summary of point-by-point response to reviewers

We thank the reviewers for their comprehensive comments on our Revised Manuscript (NCOMMS-21-06349A) and their highly constructive suggestions. A detailed point-by-point response to all the reviewer's comments are provided in this document. We performed a series of additional experiments to address the reviewers' comments, and generated new data to that effect. We believe that these additional experiments bolster the manuscript's central claims and have significantly strengthened our conclusion.

REVIEWER COMMENTS

Reviewer #1 (Remarks to the Author):

The authors provided several new data to address most of my concerns, but the key concern still remains.

1. The authors nicely provided insulin and somatostatin data in the setting of HADH overexpression. However, there must be some miscommunication. The most important thing this reviewer asked to assess is whether HADH "knockdown" mice (in Fig.1) are insulin-resistant (due to fat accumulation), which can explain all the serum glucose levels and hepatic gluconeogenesis gene expression, regardless of the authors' proposed mechanisms. Insulin serum levels do not probe hepatic insulin sensitivity. The authors should directly measure pAkt or other markers of hepatic insulin sensitivity in HADH knockdown mice.

Response: We thank the reviewer for this suggestion. We tested the protein levels of p-GSK3 β and p-AKT, markers of insulin sensitivity, in the liver of HADHA-knockdown mice. Results showed that glucagon did decrease the levels of p-GSK3 β and p-AKT, while this effect was not altered when HADHA was knocked down in mice (Supplementary Fig. 4d). In addition, HADHA knockdown had no effect on p-GSK3 β and p-AKT in normal mice (Supplementary Fig. 4d). These findings suggested that HADHA knockdown did not impair hepatic insulin sensitivity. We have added the relevant information in the Revised Manuscript.

Supplementary Fig. 4d Western blotting analysis of p-GSK3β, GSK3β, p-AKT and AKT in the liver of mice injected with AAV8-HADHA shRNA or AAV8-NC in the absence or presence of glucagon (2 mg/kg, 1 h, n = 3). AAV adeno-associated virus, GLC glucagon, NC normal control, ns no significance. Values represent mean ± SEM. Statistical difference were determined by one-way ANOVA. ***p* < 0.01 vs. the control group. Source data are provided as a Source Data file.

2. To assess the physiological relevance of BHB administration, how much BHB blood levels are elevated after exogenous BHB administration (100mg/kg) in Fig.3?

Response: To address the reviewer's comments, we tested the levels of serum BHB in the mice administrated with BHB (100 mg/kg). The absolute level of serum BHB after BHB administration is 277.6 μmol/L (Supplementary Fig. 7), 1.8-fold higher than 150.2 μmol/L in basic state. Similarly, the BHB level was also significantly elevated after BHB administration in the presence of glucagon (Supplementary Fig. 7). We have added the relevant information in the Revised Manuscript.

Supplementary Fig. 7 The serum level of BHB was elevated after BHB administration in mice. BHB (100 mg/kg) was administrated with or without glucagon (2 mg/kg, 1 h) in mice (n = 5). BHB β-hydroxybutyrate, GLC glucagon. Values represent mean ± SEM. Statistical difference were determined by one-way ANOVA. ***p* < 0.01 vs. the control group. ##*p* < 0.01 vs. indicated treatments. Source data are provided as a Source Data file.

3. It is nice to see acetyl-coA level changes as new data. Can the authors show acetyl-coA labeling from 13C-palmitate?

Response: As requested by the reviewer, acetyl-CoA was analyzed by stable isotope tracing and mass spectrometry (Supplementary Materials and Methods). [U-

^{13}C]palmitate (0.1 mM) was added to primary hepatocytes and incubated for 4 h. Analysis of both labeled (M2) and unlabeled (M0) metabolite fractions showed that glucagon stimulation slightly inhibited the level of acetyl-CoA. HADHA overexpression considerably enhanced acetyl-CoA generation, and restored acetyl-CoA production that was inhibited by glucagon (Supplementary Fig. 5a). We have added the relevant information in the Revised Manuscript.

Supplementary Fig. 5a Relative acetyl-CoA level in primary hepatocytes treated with [U- ^{13}C]palmitate (0.1 mM) for 4 h following 100 nM glucagon stimulation for 1 h with or without HADHA plasmid transfection (n = 4). GLC glucagon. Bars represent mean \pm SEM values. Statistical differences were determined by one-way ANOVA. * p < 0.05, ** p < 0.01 vs. the control group; # p < 0.05 vs. indicated treatments. Source data are provided as a Source Data file.

Supplementary Materials and Methods

Stable isotope tracing of acetyl-CoA by quadrupole time-of-flight mass spectrometry. For stable isotope tracing, the [U- ^{13}C]palmitate was conjugated with phosphate buffer saline (PBS) containing 6.02 mM bovine serum albumin (BSA). Primary hepatocytes were transfected with HADHA plasmid for 24 h and treated with [U- ^{13}C]palmitate (0.1 mM) for 4 h. For the extraction of acetyl-CoA, primary hepatocytes were first washed once with ice-cold PBS to remove the medium components and subsequently rinsed with water. Then the cells were ultrasound-crushed in 1 mL of methanol for 10 min. After centrifugation (13,000 rpm, 4 $^{\circ}\text{C}$, 10 min), 900 μL of supernatants were carefully transferred and dried by a gentle stream of nitrogen gas. Finally, the residue was reconstituted with 100 μL of 50% methanol aqueous solution, and 2 μL injection was analyzed by mass spectrometry.

Targeted detection was performed on a LC-20A system coupled to a triple quadrupole mass spectrometer (Shimadzu, UPLC-MS/MS 8050) operating in the negative ion mode. The chromatographic separation was achieved on a reversed-phase HSST3 column (Waters, 2.1×100 mm, 1.8 μm) maintained at 40 °C at a flow rate of 0.2 mL/min. The mobile phase consisted of water with 5 mM ammonium formate (A) and acetonitrile (B). The gradient elution program was 5-20% B at 0-5 min, 20-95% B at 5-8 min, 95% B at 8-10 min, and then back to initial conditions, with 2 min for equilibration. The ESI source parameters were set as follows: DL temperature, 250 °C; interface temperature, 250 °C; heat block temperature, 400 °C; heating gas flow, 10 L/min; nebulizing gas flow, 3 L/min; and drying gas flow, 10 L/min.

4. In the method, I would use 10mM instead of 40% as the BSA concentration used for tracing so that the readers can quickly notice palmitate:albumin ratio is 2:1.

Response: Thank you for the reviewer's suggestion. We have used 6.02 mM instead of 40% as the BSA concentration for tracing. This information has been detailed in the Methods in the Revised Manuscript.

Reviewer #2 (Remarks to the Author):

Most of the concerns that I raised have been properly addressed. The quality of SPR data has been improved. HDAC5's data look good. The data of HDAC4 and HDAC7 are not as good, but acceptable. However, there remain two minor points.

Response: Thank you for the reviewer's positive comments.

1. Please provide Rmax values for SPR fitting results (Fig. 6a). These values are helpful in evaluating the fitting results when compared with Chi2.

Response: Based on the reviewer's comments, we have provided the Rmax values for fitting results. Results showed that BHB exhibited favorable binding affinity to HDAC4 with Rmax = 8.86, HDAC5 with Rmax = 11.39 and HDAC7 with Rmax = 18.56. We have added the relevant values in the Fig. 6a in the Revised Manuscript.

Fig. 6a The interaction of BHB with HDACs by surface plasmon resonance. A series of BHB concentrations was injected onto the HDAC4, HDAC5 or HDAC7 biosensor surface. The frequency response and fitting curves were displayed. BHB β -hydroxybutyrate, SPR surface plasmon resonance.

2. For docking of HDAC4, please provide docking scores and make a comparison with the docking result of HDAC7.

Response: The docking scores of top 5 binding poses for BHB with HDAC4 ranged from -3.4 kcal/mol to -3.67 kcal/mol, which is similar to BHB with HDAC7 (-3.55 ~ -3.66 kcal/mol). In a series of subsequent biological validation, we found that HDAC7, but not HDAC4 and 5, was restrained in the cytosol by HADHA overexpression in hepatocytes (Fig. 5a). Immunoprecipitation assays showed that HADHA overexpression or BHB addition visibly reduced the binding of HDAC7 to FOXO1, but had no significant influence on the binding of HDAC4 and 5 to FOXO1 (Fig. 5b, Supplementary Fig. 13d). Glucagon increased enzymatic activities of HDACs, whereas BHB considerably inactivated HDAC7 without significant influence on HDAC4 and HDAC5 activity (Fig. 5c). Taken together, BHB selectively inhibited HDAC7 to preserve FOXO1 acetylation.

Reviewers' Comments:

Reviewer #1:

Remarks to the Author:

No more concerns.

Reviewer #2:

Remarks to the Author:

All the concerns that I raised have been properly addressed.